# Automatic high-frequency measurements of full soil greenhouse gas fluxes in a tropical forest

Courtois Elodie A. [1,2,*,#], Stahl Clément [3,#], Burban Benoit [3], Van den Berge Joke [1], Berveiller Daniel [4], Bréchet Laëtitia [1,3], Soong Jennifer L.[1,5], Arriga Nicola [1], Peñuelas Josep [6,7], Janssens Ivan A. [1]

[1] Centers of Excellence Global Change Ecology and PLECO (Plants and Ecosystems), Department of Biology University of Antwerp, Universiteitsplein 1, B-2610 Wilrijk, Belgium

[2] Laboratoire Ecologie, évolution, interactions des systèmes amazoniens (LEEISA), Université de Guyane, CNRS, IFREMER, 97300 Cayenne, French Guiana.

[3] INRA, UMR EcoFoG, CNRS, Cirad, AgroParisTech, Université des Antilles, Université de Guyane, 97310 Kourou, France

[4] CNRS/Université Paris-sud, 362 rue du doyen André Guinier, 91405 Orsay Cedex

[5] Climate and Ecosystem Sciences Division, Lawrence Berkeley National Laboratory, 94720, Berkeley, USA

[6] CSIC, Global Ecology Unit CREAF-CSIC-UAB, Bellaterra, 08193 Catalonia, Spain

[7] CREAF, Cerdanyola del Vallès, 08193 Catalonia, Spain

*Correspondence to*: Elodie A. Courtois (elodie.courtois@cnrs.fr~~courtoiselodie@gmail.com~~)
# the two first authors contributed equally to the manuscript.

**Abstract.** Measuring *in situ* soil fluxes of carbon dioxide ($CO_2$), methane ($CH_4$), and nitrous oxide ($N_2O$) continuously at high frequency requires appropriate technology. We tested the combination of a commercial automated soil $CO_2$ flux chamber system (LI-8100A) with a $CH_4$ and $N_2O$ analyzer (Picarro G2308) in a tropical rainforest for 4 months. A chamber closure time of 2 minutes was sufficient for a reliable estimation of $CO_2$ and $CH_4$ fluxes (100% and 98.5% of fluxes were above Minimum Detectable Flux – MDF, respectively). This closure time was generally not suitable for a reliable estimation of the low $N_2O$ fluxes in this ecosystem but was sufficient for detecting rare major peak events. A closure time of 25 minutes was more appropriate for reliable estimation of most $N_2O$ fluxes (85.6% of measured fluxes are above MDF $\pm$ 0.002 nmol m$^{-2}$ s$^{-1}$). Our study highlights the importance of adjusted closure time for each gas.

## 1 Introduction

After water vapour, ~~C~~carbon dioxide ($CO_2$), methane ($CH_4$) and nitrous oxide ($N_2O$) are the three main greenhouse gases (GHGs) in terms of radiative forcing. Increases in these GHG concentrations in the atmosphere is driving anthropogenic global warming. Understanding the magnitude of GHG fluxes in natural ecosystems has recently become a priority in the study of GHG balances (Merbold et al., 2015). Tropical intact forests cover 1392 Mha globally and represent about 70% of the total tropical forest area (1949 Mha), which accounts for the largest area of global forest biomes (~50%). Very few reliable long term datasets on full GHG balances are available from tropical ecosystems, despite their known importance for the global cycles of these three GHGs (Dutaur and Verchot, 2007). This is in part due to the challenges of designing and operating continuous, multi-gas flux analysis systems in tropical forests. Soil processes in particular are responsible for an important part

of GHGs that are produced or consumed in tropical ecosystems (Oertel et al., 2016). Soil physical, chemical, and biological characteristics are linked to variation in GHG emissions from soils, which in turn can display very high spatial and temporal variability (Arias-Navarro et al., 2017; Silver et al., 1999).

Historically, soil GHG fluxes (emission or consumption) have been measured using the static chamber method. This involves closing chambers manually for a known period of time, usually 30-60 minutes, and repeated collection of air samples for further analysis via gas chromatography (Verchot et al., 1999, 2000). Fluxes are then computed from the ~~increase (or decrease)~~change in gas concentration per unit time, per surface area enclosed by the chamber, and corrected by the volume of the chamber. While these labor-intensive and time-consuming manual measurements are well adapted to capture high spatial flux variability (Arias-Navarro et al., 2017; Pumpanen et al., 2004), they do not capture high temporal variation, which is necessary for the accurate estimation of annual GHG budgets. Moreover, short term, transient spikes in the emission or consumption of these GHGs likely remains undetected with static chamber methods, imposing a lost opportunity to fully understand the production or consumption processes of GHGs and their response to rapidly changing environmental conditions. One of the key challenges of contemporary GHG flux research is to close these knowledge gaps in order to improve the quantitative prediction of GHG fluxes (Merbold et al., 2015).

The use of automatic chambers is one approach to obtain continuous estimation of soil GHG flux data at high temporal frequency (several measurements per days) at various sampling points. Since the 1970s (Denmead, 1979), a variety of technical solutions for automated flux sampling have been developed (Ambus et al., 2010; Breuer et al., 2000; Görres et al., 2016; Kostyanovsky et al., 2018; O'Connell et al., 2018; Petrakis et al., 2017a; Savage et al., 2014), particularly for soil $CO_2$ fluxes. However, accurate detection of $CH_4$ and $N_2O$ fluxes from soils via flow through systems is more difficult than $CO_2$ due to significantly lower background concentrations and lower flux rates (Kostyanovsky et al., 2018). The budgetary requirements for large infrastructure and intensive maintenance as compared to manual chamber measurements have prevented the widespread application of automated systems. The use of automated and continuous methods to estimate full GHG budgets *in situ* remains scarce, especially in complex biomes with extreme climate such as tropical forests. Therefore, only a few studies actually address the difficulties and challenges associated with operating these systems under field conditions (Görres et al., 2016; Koskinen et al., 2014).

Recent technological advances have now made more automated chamber systems commercially available, and an increasing number of custom-made systems are being designed and deployed for soil GHG flux measurements (De Klein and Harvey, 2012). Here, we present a detailed field deployment of a custom built, automated soil GHG flux system – the LI-8100A Soil $CO_2$ Flux System (LI-COR Biosciences Inc., Lincoln, NE, USA) running in line with a Picarro G2308 (Picarro Inc., Santa Clara, CA, USA). Using a 4 month dataset of continuous measurements of $CO_2$, $CH_4$, and $N_2O$ fluxes simultaneously under tropical forest conditions, we present an optimized sampling protocol for the estimation of the full GHG budget in this ecosystem.

## 2 Methods

### 2.1 Measurement site

This study was conducted in the Paracou research station (5°15'N, 52°55'W), located in the coastal area of French Guiana, South America. The automated soil GHG flux system was deployed in the footprint of the Guyaflux site, which holds a 55 m-tall tower upon which canopy $CO_2$, $H_2O$ and energy fluxes have been monitored since 2004 using the eddy covariance technique (Aguilos et al., 2018; Bonal et al., 2008). The site is covered with tropical pristine forest and located in the northernmost part of the Guiana shield. It is characterized by a succession of small, elliptical hills rising to 10–40 m a.s.l., sometimes associated with plateaus of similar altitude.

The soils are mostly nutrient-poor acrisols (FAO-ISRICISSS, 1998) with pockets of sandy ultisols developed over a Precambrian metamorphic formation called the 'Bonidoro series', and composed of schist and sandstone, sporadically traversed by veins of pegmatite, aplite and quartz (Bonal et al., 2008). The forest around the tower is characteristic of a tropical pristine forest with both high tree density (~ 620 trees with a dbh>10 cm ha$^{-1}$) and species richness (~ 140 species ha$^{-1}$). The climate is highly seasonal due to the north/south movement of the Inter-Tropical Convergence Zone. The wet season, characterized by heavy rain events, lasts for 8 months (December–July) and alternates with a 4 month dry period (August–November) during which precipitation is typically lower than 100 mm per month. For the period 2004-2015, annual rainfall quantities were on average 3103 mm year$^{-1}$, relative extractable water (an index of soil water availability (Wagner et al., 2011)) varied from 0.93 in the wet season to 0.46 in the dry season and soil temperature was on average 25.1 with little seasonal nor diurnal variation (Aguilos et al., 2018).

### 2.2 Automated sampling system

A schematic view of the automatic sampling system is shown in Figure 1(A). The system consisted of four main components: sixteen automated long-term chambers (8100-104, LI-COR Biosciences), a multiplexer to link one chamber at a time to the gas analyzers (LI-8150, LI-COR Biosciences), an infrared gas analyzer (IRGA) to measure $CO_2$ concentrations (LI-8100A, LI-COR Biosciences), and a cavity ring down spectroscopy (CRDS) instrument to measure $CH_4$ and $N_2O$ concentrations (G2308, Picarro) that was fitted with an external recirculation pump (A0702, Picarro). Both the IRGA and CRDS systems were necessary to measure all three GHG concentrations due to the different abundances and flux rates of $CO_2$, $CH_4$ and $N_2O$. The IRGA methodology is accurate and precise enough to detect small $CO_2$ concentration changes at high background concentrations (approximately 400 ppmv; parts per million in volume units). However, the detection of small changes in $CH_4$ and $N_2O$ concentrations, even at their low background atmospheric concentrations in the order of 2000 ppbv (ppbv=parts per billion in volume units) and 300 ppbv, respectively, requires higher accuracy and precision levels that can be detected with the CRDS.

Power supply was delivered through a 12 kVa generator (Perkins STORM15) fitted with batteries located 400 m away from the instruments. Both the $CO_2$ analyzer control unit and the multiplexer (LI-COR) had their own weather-proof casing,

requiring no additional protection in the field. Nonetheless, in consideration of the high precipitation at the site, these devices were placed under a wooden shelter for added protection. The $CH_4$ and $N_2O$ analyzer (Picarro), its external pump and a computer monitor were housed in a waterproof shelter that was specifically designed to host them (Figure 1(C)). The LI-8100 and the G2308 computers were connected through ethernet connection to ensure time synchronization. The sixteen automated

soil chambers (8100-104, LI-COR Biosciences) were installed in a grid in the forest (Figure 1(B)) covering in total an area of approximately 300 m$^2$ (15 m x 20 m). Each chamber was only closed during individual chamber measurement periods, and was fully open when not sampling. The PVC collars that were provided with the 8100-104 automatic chambers were inserted in the soil one month prior to the first measurement (20.3 cm inner diameter/21.3 cm outer diameter; enclosed soil area ~ 318 cm$^2$; insertion depth ~ 7cm; offset ~ 4cm; green PVC). When the chambers close, they are automatically lowered so that they

cover each soil collar and ensure a fully sealed chamber. The chamber lid does not directly rest on the collar rim, but on a metal plate surrounding the collar, leaving the collar undisturbed and minimizing lateral leaks (Hupp et al., 2009).

The 16 chambers were connected via 15 m Bev-a-line tubing (8 mm inner diameter) with the multiplexer (LI-8150), which allows for switching between each of the 16 chambers in any given sequence. Soil temperature (at a depth of 10 cm) was monitored with 8100-201 Ω thermistor probes (Omega Engineering Inc., Stamford, CT, USA), and soil volumetric water

content (0-10 cm) was monitored with 8100-202 ECH$_2$O Model EC-5 soil moisture sensors (Decagon Devices Inc., Pullman, WA, USA). Soil temperature and soil volumetric water content sensors were directly connected to the chambers and recorded by the Licor system using the same time step.

Each chamber was purged for 15 sec prior to each measurement and 45 sec after each measurement in order to flush the lines and restore background gas levels in the system. The flow rate during the purging and the measurements was ~2.8 L min$^{-1}$

between the LI-8150 and the chambers, which ensures sufficient air mixing in the chamber headspace during the measurements (Görres et al., 2016). Flow rates in the subsampling lines (Li8100 and Picarro) were lower and set between 1.5 and 1.7 L min$^{-1}$ as recommended by the manufacturers. The LI-8100 software provided the rate of $CO_2$ concentration increase in the chamber which was used to quantify the flux of $CO_2$ from the soil surface into the atmosphere (taking into account the enclosed soil surface area and the total system volume). A subsampling loop was inserted after the analyzer (LI-8100A) and

before the multiplexer (LI-8150), to pull the air sample through the Picarro G2308 CRDS analyzer for the determination of $CH_4$ and $N_2O$ concentrations and flux estimations, before going back to the chamber (Figure 1(A)). All three gas concentrations were recorded every second over the sampling periods.

### 2.3 Flux calculations

All fluxes estimation were done by using commercially available Soil Flux pro software (LI-COR Biosciences). An R script (Supplementary file 1) was created to merge all the Picarro files from a given week in order to import them into the Soil Flux Pro software. The Picarro creates one file per hour and when Picarro files are not merged, Soil Flux Pro software is not able to deal with measurements overlapping between two distinct Picarro files (e.g. when a single measurement is done from 9:50 am to 10:15 am) leading to incorrect estimation of $CH_4$ and $N_2O$ fluxes. To avoid underestimation of fluxes (Supplementary

Figure 1), CO₂, CH₄ and N₂O fluxes were measured as exponential fit of gas concentration with time using Soil Fflux Pro software and include a 60 sec dead band to account for soil surface pressure disturbances due to the closing of the chamber.

## 2.4 Minimum Detectable Fluxes

The minimum detectable flux (MDF) for each gas was estimated by using a metric originally developed by Christiansen et al. (2015), which was modified by Nickerson (2016) to make it more suitable for high-frequency measurements (Christiansen et al., 2015; Nickerson, 2016):

$$MDF = \left(\frac{A_a}{t_c\sqrt{n}}\right)\left(\frac{VP}{SRT}\right)$$

Where Aa is the analytical accuracy of the analyzer (25 ppb for N₂O and 10 ppb for CH₄ with the Picarro G2308 and 600 ppb for CO₂ with the Li8100, recorded from the technical data sheets of the analyzers), $t_c$ is the closure time of the chamber in seconds, $n$ is the number of points that are available to compute the flux (i.e. $t_c$ divided by the sampling periodicity, every 1 second in this study), V is the chamber volume (0.0040761 m³), P is the atmospheric pressure (101325 Pa), S is the chamber surface area (0.03178 m²), R is the ideal gas constant (8.314 m³ Pa K⁻¹ mol⁻¹) and T is the ambient temperature (298.15 K). We computed the MDF of each gas for closure times from 2 minutes to 30 minutes in order to select the optimal chamber closure time for each gas in our integrated system (Table 1).

## 2.5 Closure time

Selecting the best length of time for soil GHG measurements and accurate flux calculation in an integrated CO₂, CH₄ and N₂O automated measurement system requires careful consideration. At low fluxes, longer measurement periods are needed to reach reliable measurements of real concentration changes, while at high fluxes possible storage and saturation effects in the chamber headspace might result in non-linear concentration increases and thereby underestimated fluxes if fluxes are calculated linearly. In order to maximize the detectable percentage of fluxes for N₂O and CH₄ without impeding spatial coverage and temporal resolution, we built a combined program with two different closure times. Each week, four out of sixteen chambers were programmed to stay closed for a longer measurement period to ensure a reliable estimation of low fluxes while the other twelve chambers were programmed to stay closed for a shorter period to capture diel variation and detect high fluxes. For the short closure time (SHORT hereafter), we used a 2 minute measurement period because (1) this is a standard closure time for soil CO₂ flux calculations (Janssens et al., 2000), even in tropical forests (Epron et al., 2006; Sayer et al., 2007) of this region where MDF for CO₂ flux is typically low (Bonal et al., 2008; Bréchet et al., 2009; Courtois et al., 2018), (2) corresponding MDFs of CH₄ (0.04 nmol m⁻² s⁻¹ or ) and N₂O (0.1 nmol m⁻² s⁻¹) are compatible with the detection of emission or consumption peaks of these two gases in this region (Courtois et al., 2018; Petitjean et al., 2015). For the long closure time (LONG hereafter), we decided to use a 25 minute measurement period in order to optimize the trade-off between a reliable estimation of low N₂O fluxes (Table 1) and a program length that allows for a sufficient number of flux measurements per chamber and per day.

We therefore programmed the multiplexer for 2.5-h cycles (9-10 measurements per chamber per day), which included four chambers with LONG measurements and twelve chambers with SHORT measurements. Each week, the program was modified manually so that the four LONG measurements were rotated across the chambers. Each chamber was therefore measured with the LONG closure time for one 7 consecutive day period per month (4 weeks).

### 2.6 System maintenance and data processing

The automated sampling system was installed on June 1[st] 2016 and operated until September 29[th] 2016 (4 months), totaling 17592 individual measurements for each gas (4098 with LONG closure time and 13494 with SHORT closure time). Coarse wood debris were removed weekly but small litter, such as leaves, fruits, and twigs, was left in the collar area. Every week,

living plants growing inside the collars, and the dead leaves on the chambers, were carefully removed by hand. The $R^2$ value of the exponential increase of $CO_2$ over 2 minutes was used as an indicator that the system was functioning correctly and not impeded by debris (Görres et al., 2016; Savage et al., 2014). When the $R^2$ of the regression between time and $CO_2$ concentration was lower than 0.9, we considered this as an indication that there may have been an issue with the chamber closing and sealing correctly and removed the flux measurement for all three gases from our analysis.

For $CO_2$, we observed a strong concentration saturation effect when using the LONG closure time (25 minutes), leading to an underestimation of fluxes (Figure 2). All $CO_2$ flux estimates were therefore based on 2 minute regressions only, using either full concentration measurements of the SHORT closure time or the 2 first minutes of the LONG closure time. Following recommendations (Rubio and Detto, 2017), we removed anomalous values, i.e. $CO_2$ fluxes estimation with a difference greater than 5 µmol m$^{-2}$ s$^{-1}$ between with ~~previous or followin~~adjacent ~~g~~ measurements or lower than 0 µmol m$^{-2}$ s$^{-1}$. For $CH_4$, we

observed only a slight saturation effect when using the LONG closure time (Figure 2). Variation in the flux calculations did not differ between the SHORT and LONG chamber closure measurements. $N_2O$ flux calculations were much more variable when measuring with the SHORT closure time compared to the LONG closure time (Figure 2). Even if fluxes were above the detection limit, the low fluxes estimated with the SHORT closure time were not reliable as shown by the low correlation in Figure 2. For both $CH_4$ and $N_2O$, we therefore decided to apply the following quality check procedure (1) All fluxes that were

not complying with MDF criterion were discarded. (2) All fluxes estimated with the SHORT closure time with a $R^2$ lower than 0.8 were discarded (Savage et al., 2014). (3) We applied the same procedure than for $CO_2$ regarding anomalous values (difference greater than 5 nmol m$^{-2}$ s$^{-1}$ between consecutive measurements).

### 3 Results and discussions

A cleaning frequency of once a week was necessary and sufficient to remove falling leaves and branches from the automatic chamber system, prevent leaks and generate a continuous dataset of soil GHG fluxes from this tropical forest. Temperature variations are typically small below the canopy due to the shadowing by dense canopy crown and microclimatic conditions. During the study period, temperature at 2m height varie~~s during the day~~ds from 22 °C in the night to 28 °C during the day. The

presence of water condensation inside the tubing lines was carefully checked every week and never occurred during the study period. The automatic chamber system worked well most of the time, but some data gaps did exist. Over the 17592 individual flux estimations, 343 (1.9 %) had to be discarded because of (1) problems in the connection between the chamber and the multiplexer (154 measurements, 0.9% of data points); (2) imperfect chamber closing, which was detected by an insufficient

increase of $CO_2$ (189 measurements, 1% of data points).

### 3.1 $CO_2$ fluxes

Additionally to the 343 fluxes than were removed after the firsts steps of quality check procedure, 758 $CO_2$ fluxes estimations were also considered as anomalous, either because the difference with previous or following measurements where greater than

5 µmol m$^{-2}$ s$^{-1}$ (758 measurements, i.e. 4.3%) or because they were lower than 0 µmol m$^{-2}$ s$^{-1}$ (14 measurements). In total, 16477 $CO_2$ fluxes over 17592 (93.6%) ~~can~~ could be used over the four month~~s~~ period. $CO_2$ fluxes were on average 8.1~~06~~ ± 1.6 µmol m$^{-2}$ s$^{-1}$(Table 2) which would correspond to a mean annual soil $CO_2$ efflux of 3050 gC m$^{-2}$ year$^{-1}$ which falls into the upper range of the extensive review of mean annual soil $CO_2$ effluxes estimations in tropical forest provided recently by Rubio and Detto (2017). Nonetheless, our study period (June-September) only covered the end of the wet season and more data are

needed to precise this estimation. . All two-minute measurements of $CO_2$ fluxes from the four-month study period were above the MDF of 2.39 nmol m$^{-2}$ s$^{-1}$ for the LI8100 analyzer (Table 1). No saturation effect was detected using the SHORT closure time and estimation of $CO_2$ over a shorter time period is not recommended (Davidson et al., 2002). $CO_2$ fluxes using the LONG closure time would be underestimated due to the buildup of high $CO_2$ concentrations due to large fluxes over this long time period (Figure 2), and are not recommended. For small chambers as the one that were used in this study, we therefore conclude

that a 2 minute sampling time should be used for $CO_2$ flux calculations since the MDF of this short measurement period allowed for the retention of 100% of the data. When the chambers stay closed longer for accurate detection of $N_2O$ fluxes, only the first two minutes of data should be used for $CO_2$ flux calculations.

The use of 16 automated flux chambers allowed for the capture of spatial and temporal variability of soil respiration. Over this four month period, corresponding to the end of the wet season in French Guiana, temporal variability remained low (Figure

4). This dataset is therefore not long enough to detect seasonal variation of soil respiration that were highlighted in previous study (Rowland et al., 2014; Rubio and Detto, 2017). We did found that soil respiration tended to decrease in very humid soils (Supplementary Figure 2~~1~~) as highlighted previously at the same site (Rowland et al., 2014) but more data are needed to disentangle precisely the importance of seasonal and diurnal variability from the responses to environmental triggers on soil respiration. Nonetheless, even during this relatively short period, our data clearly demonstrated a strong spatial variability of

soil respiration, even at a low spatial scale (Figure 5, Table 2), some local spots clearly displaying stronger values of soil respiration during the study period.

## 3.2  CH₄ fluxes

Additionally to the 343 fluxes than were removed after the firsts steps of quality check procedure, CH$_4$ fluxes estimations were also discarded because of (1) problems with Picarro files (12 measurements), (2) application of the MDF criterion (137 measurements), (3) application of the R$^2$ criterion for SHORT closure time (3751 measurements, i.e. 28% of the SHORT measurements) and (4) detection of anomalous values (364 measurements). In total, 12985 CH$_4$ fluxes over 17592 (73.8%) ~~can~~ could be used over the four months period. No saturation effect was detected using the LONG closure time and fluxes estimated with the SHORT closure time were very well correlated to fluxes using the LONG closure time, even for small fluxes (Figure 2). 68.4 % and 98.2% of fluxes measured with the SHORT and LONG closure times, respectively, were retained in our quality control data processing over the four-month study period. These measurement periods, therefore, allowed for the retention of a large majority of CH$_4$ emission or consumption fluxes in our data analysis.

CH$_4$ fluxes were on average 1.7 ± 3.8 nmol m$^{-2}$ s$^{-1}$ with a high variability among chambers (Table 2) but the frequency of negative CH$_4$ fluxes (consumption, 59% of fluxes) was greater than positive fluxes (emission, 41% of fluxes) during this period (Figure 3). Most of the time, soils were either consuming or emitting small amounts of CH$_4$, but transient, large emission peaks were periodically detected at individual chamber locations during the study period (Figure 6). Tropical soils are generally considered as sink at a yearly basis (Dutaur and Verchot, 2007) but it is known that these soils can shift from a source in the wet to a sink in the dry season (Courtois et al., 2018; Davidson et al., 2008; Teh et al., 2014). No clear temporal trend could be detected during the study period and no clear pattern linked CH$_4$ fluxes with surface soil humidity (Supplementary Figure 2~~1~~), but as for CO$_2$, longer time series covering at least a full year are needed to explore the seasonal and diurnal variability of fluxes. As highlighted previously in French Guiana (Courtois et al., 2018), spatial variability of CH4 emission was high, even at a small spatial scale (Figure 5, Figure 6). Interestingly, some spots clearly display~~ed~~ high CH$_4$ emission during all the study period (Figure 5, Figure 6).

## 3.3  N₂O fluxes

Additionally to the 343 fluxes than were removed after the firsts steps of quality check procedure, N$_2$O fluxes estimations were also discarded because of (1) problems with Picarro files (12 measurements), (2) application of the MDF criterion (1594 measurements), (3) application of the R$^2$ criterion for SHORT closure time (11643 measurements, i.e. 28% of the SHORT measurements) and (4) detection of anomalous values (364 measurements). In total, 3998 N$_2$O fluxes over 17592 (22.7%, 140 measurements with the SHORT and 3858 measurements with the LONG closure time) ~~can~~ could be used over the four months period. 94.1% of fluxes measured with the LONG closure times were retained after our quality control data processing over the four-month study period. When measured over 25 minutes, N$_2$O fluxes in our site could therefore be considered as reliable. Using the SHORT closure time, most flux estimations had to be discarded because they led to unreliable flux estimations (Figure 2). Nonetheless, the SHORT closure time still allowed the detection of high N$_2$O emission or consumption events tha~~t~~n were detected during the study period (Figure 5 and 7).

N$_2$O fluxes were on average 0.1 ± 0.2 nmol m$^{-2}$ s$^{-1}$ with a high variability among chambers (Table 2). At the same chamber, N2O flux can shift from consumption to emission with 28% of fluxes indicating a sink and 72% a source for N$_2$O (Figure 3). The high variability in N$_2$O fluxes that we detected over four months with our automated system are in agreement with the typical high variability in N$_2$O fluxes measured from tropical soils over space and time using static chambers (Arias-Navarro et al., 2017; Courtois et al., 2018). Moreover, N$_2$O fluxes didn't show any relationship with surface soil humidity (Supplementary Figure 2~~1~~), which underline the complexity of the biological process underlying these fluxes. In a previous study in the same environment (Courtois et al., 2018), we estimated that the minimum detectable fluxes using Gas Chromatography analysis of four discrete gas samples over 30 minutes for N$_2$O was ± 8.3 µg N m$^{-2}$ h$^{-1}$. MDF estimated in the present study using high frequency measurement was 0.002 nmol m$^{-2}$ s$^{-1}$ or 0.2 µg N m$^{-2}$ h$^{-1}$ for N$_2$O which is therefore ~ 40 times lower. Such result indicates that this long-term system is well-adapted to capture and estimate the low N$_2$O fluxes occurring in this ecosystem.

## 4    Conclusions

~~Our system coupled a Li8100 CO$_2$ analyzer and multiplexor with a Picarro G2308 CH$_4$ and N$_2$O analyzer to sample 16 automated soil flux chambers with a rotation of SHORT and LONG closure times for the accurate monitoring of three GHG fluxes over four months with high spatial and temporal resolution. The sampling system of SHORT and LONG closure times with a weekly rotation presented here has three major advantages, which ultimately can provide high confidence in the estimation of annual the full GHG budgets of tropical soils: (1) the LONG closure time allows a reliable estimation of the low N$_2$O fluxes in this ecosystem, which was clearly not achieved using a shorter closure time, (2) the number of data points per day are sufficiently high (9 to 10 measurements per day) to capture potential diurnal variation (Nicolini et al., 2013; Rubio and Detto, 2017) of the three gases with good spatial replication (16 chambers), (3) periodic extreme events of high N$_2$O fluxes can still be detected with the SHORT closure time period, which occurs at higher frequency than the LONG closure measurements. Our study underlines the importance of appropriate closure time for each GHG gas for accurate estimation of GHG budgets.~~

We demonstrated here that the combination of a commercial soil GHG chamber system – the LI-8100A Automated Soil CO$_2$ Flux System – running in line with a Picarro G2308, enables the continuous, long-term measurement of CO$_2$, CH$_4$, and N$_2$O simultaneously under tropical conditions. Similar configurations have been recently implemented in temperate climate (Petrakis et al., 2017b, 2017a), but to our knowledge, this is the first time that this experimental set up is fully described and tested under tropical field conditions for the measurement of the three soil GHG fluxes simultaneously~~-~~. Additionally, our study determined the optimal chamber closure time for each GHG. The sampling system of SHORT and LONG closure times with a weekly rotation presented here has three major advantages, which ultimately can provide high confidence in the estimation of annual the full GHG budgets of tropical soils: (1) the LONG closure time allows a reliable estimation of the low N$_2$O fluxes in this ecosystem, which was clearly not achieved using a shorter closure time, (2) the number of data points per

day are sufficiently high (9 to 10 measurements per day) to capture potential diurnal variation (Nicolini et al., 2013; Rubio and Detto, 2017) of the three gases with good spatial replication (16 chambers), (3) periodic extreme events of high $N_2O$ fluxes can still be detected with the SHORT closure time period, which occurs at higher frequency than the LONG closure measurements. Our study underlines the importance of appropriate closure time for each GHG gas for accurate estimation of GHG budgets. This information is crucial for the calculation of accurate soil fluxes at diurnal timesteps and for the estimation of annual GHG budgets. This combination of automated closed dynamic chambers and advanced GHG analyzers allows for, (1) accounting of short-term variability in GHG fluxes while taking into account spatial variability, (2) estimating annual GHG budgets at these locations, (3) tracking the variability in GHG fluxes along hours, days, seasons and years, and (4) studying the impact of climatic change on soil GHG budgets.

**Author contribution.** JVB and NA designed the experiments and EAC, CS, BB and DB carried them out. EAC and CS prepared the manuscript with contributions from all co-authors.

**Competing interests.** The authors declare that they have no conflict of interest.

**Acknowledgments.** This research was supported by the European Research Council Synergy grant ERC-2013-SyG 610028-IMBALANCE-P. We thank J. Segers for help in the initial setting of the system and R. Winkler from Picarro and R. Madsen and J. Hupp from LI-COR for their help in combining the systems. We thank the staff of Paracou station, managed by UMR Ecofog (CIRAD, INRA; Kourou) which received support from "Investissement d'Avenir" grants managed by Agence Nationale de la Recherche (CEBA: ANR-10-LABX-25-01, ANAEE-France: ANR-11-INBS-0001). The Guyaflux program belongs to the SOERE F-ORE-T which is supported annually by Ecofor, Allenvi and the French national research infrastructure, ANAEE-F. This program also received support from an "investissement d'avenir" grant from the Agence Nationale de la Recherche (CEBA, ref ANR-10-LABX-25-01). IAJ acknowledges support from Antwerp University (Methusalem funding), NA from ICOS-Belgium and Fonds Wetenschappelijk Onderzoek (FWO) and JLS from the U.S. Department of Energy funding under contract DE-AC02-05CH11231.

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

**Table 1:** Minimum Detectable Fluxes (MDF) for each gas and for closure times from 2 to 30 minutes. The two closure times that were used in this study (2 minutes and 25 minutes) are highlighted in bold.

| Closure time (minutes) | $N_2O$ (nmol m$^{-2}$ s$^{-1}$) | $CH_4$ (nmol m$^{-2}$ s$^{-1}$) | $CO_2$ (nmol m$^{-2}$ s$^{-1}$) |
|:---:|:---:|:---:|:---:|
| **2** | 0.100 | **0.040** | **2.393** |
| 5 | 0.025 | 0.010 | 0.605 |
| 10 | 0.009 | 0.004 | 0.214 |
| 15 | 0.005 | 0.002 | 0.117 |
| 20 | 0.003 | 0.001 | 0.076 |
| **25** | **0.002** | **0.001** | 0.054 |
| 30 | 0.002 | 0.001 | 0.041 |

**Table 2:** Mean, standard deviation (SD), minimum (Min) and maximum (Max) values of each gas and each chamber over the study period. These values are computed using all fluxes estimation (either with SHORT or LONG closure time) remaining after quality check. The number (N) of fluxes that were used is also indicated for each chamber. The last line of the table is the mean of all fluxes by chambers by gas and the min and max for all chambers by gas.

| | $CO_2$ ($\mu$mol m$^{-2}$ s$^{-1}$) | | | | | $CH_4$ (nmol m$^{-2}$ s$^{-1}$) | | | | | $N_2O$ (nmol m$^{-2}$ s$^{-1}$) | | | | |
|---|---|---|---|---|---|---|---|---|---|---|---|---|---|---|---|
| | Mean | Sd | Min | Max | N | Mean | Sd | Min | Max | N | Mean | Sd | Min | Max | N |
| Chamber 1 | 7.19 | 0.93 | 2.14 | 10.81 | 940 | 10.97 | 7.73 | -2.08 | 28.79 | 840 | 0.10 | 0.12 | -0.48 | 0.70 | 284 |
| Chamber 2 | 7.60 | 1.11 | 4.00 | 12.21 | 1166 | -1.62 | 1.75 | -4.09 | 11.68 | 899 | 0.00 | 0.14 | -1.03 | 0.75 | 285 |
| Chamber 3 | 5.58 | 0.99 | 2.11 | 11.12 | 1135 | 0.35 | 2.95 | -2.48 | 22.94 | 745 | 0.03 | 0.23 | -0.61 | 2.85 | 208 |
| Chamber 4 | 7.94 | 1.37 | 4.36 | 12.13 | 1154 | -1.85 | 1.23 | -3.63 | 6.09 | 1105 | 0.04 | 0.10 | -0.66 | 0.60 | 224 |
| Chamber 5 | 4.14 | 0.92 | 0.53 | 10.05 | 1139 | 1.37 | 3.26 | -2.20 | 12.61 | 752 | 0.15 | 0.33 | -1.04 | 3.23 | 382 |
| Chamber 6 | 8.87 | 1.70 | 3.36 | 17.68 | 1070 | -1.38 | 1.78 | -3.20 | 8.04 | 801 | -0.02 | 0.12 | -1.04 | 0.63 | 272 |
| Chamber 7 | 13.47 | 2.78 | 0.89 | 22.12 | 988 | 1.37 | 3.60 | -2.63 | 19.56 | 749 | 0.64 | 1.37 | -0.85 | 7.93 | 216 |
| Chamber 8 | 7.44 | 1.19 | 2.03 | 11.02 | 1099 | 0.03 | 2.96 | -3.37 | 18.47 | 785 | 0.02 | 0.15 | -1.36 | 0.84 | 202 |
| Chamber 9 | 4.25 | 1.20 | 0.44 | 11.37 | 1002 | 2.06 | 3.13 | -2.14 | 11.53 | 879 | 0.02 | 0.11 | -0.62 | 0.58 | 332 |
| Chamber 10 | 5.60 | 1.30 | 0.69 | 13.13 | 1037 | 1.21 | 2.46 | -1.91 | 10.34 | 657 | 0.04 | 0.13 | -0.64 | 0.77 | 252 |
| Chamber 11 | 11.97 | 2.19 | 6.84 | 18.78 | 1004 | 6.72 | 7.61 | -1.06 | 41.49 | 855 | 0.03 | 0.17 | -1.01 | 1.04 | 199 |
| Chamber 12 | 9.42 | 2.70 | 3.45 | 21.54 | 968 | 1.40 | 6.68 | -3.29 | 41.94 | 891 | 0.02 | 0.09 | -0.75 | 0.30 | 204 |
| Chamber 13 | 5.85 | 1.34 | 0.42 | 8.49 | 944 | 5.29 | 5.92 | -4.60 | 26.64 | 654 | 0.10 | 0.19 | -0.84 | 1.71 | 335 |
| Chamber 14 | 5.66 | 1.15 | 0.72 | 10.72 | 987 | 2.78 | 6.22 | -2.48 | 35.15 | 691 | 0.09 | 0.17 | -0.63 | 0.93 | 231 |
| Chamber 15 | 16.63 | 3.27 | 9.42 | 29.64 | 850 | -0.46 | 2.05 | -3.25 | 8.26 | 839 | -0.02 | 0.16 | -0.96 | 0.72 | 185 |
| Chamber 16 | 7.35 | 1.13 | 3.98 | 11.37 | 994 | -1.34 | 1.48 | -3.60 | 6.11 | 843 | 0.00 | 0.11 | -1.00 | 0.83 | 187 |
| | **8.06** | **1.58** | **0.42** | **29.64** | **16477** | **1.68** | **3.80** | **-4.60** | **41.94** | **12985** | **0.08** | **0.23** | **-1.36** | **7.93** | **3998** |

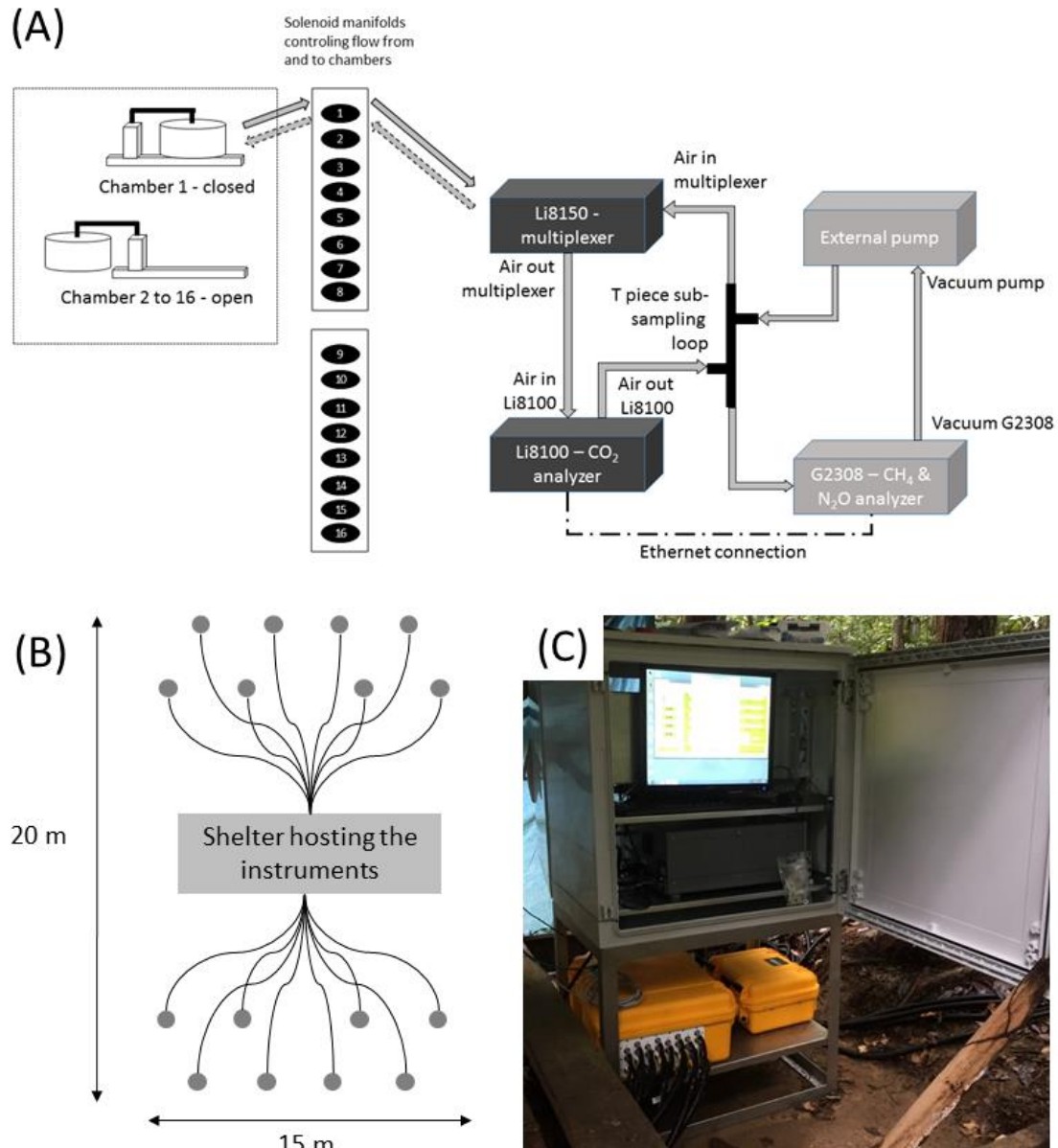

**Figure 1: Experimental Design:** (A) Schematic view of the installation composed of four main components: sixteen automated long-term chambers (8100-104, LI-COR Biosciences), a multiplexer to link one of these chambers to the gas analyzers (LI-8150, LI-COR Biosciences), an infrared gas analyzer (IRGA) to measure $CO_2$ concentrations (LI-8100A, LI-COR Biosciences), and a cavity ring down spectroscopy (CRDS) instrument to measure $CH_4$ and $N_2O$ concentrations (G2308, Picarro) that was fitted with an external pump. (B) Schematic representation of the grid with the shelter housing the equipment in the middle and the 16 chambers (grey dots) linked to the LIi-8150 multiplexer with 15 meters cables (black lines). (C) Picture of the instruments in the field.

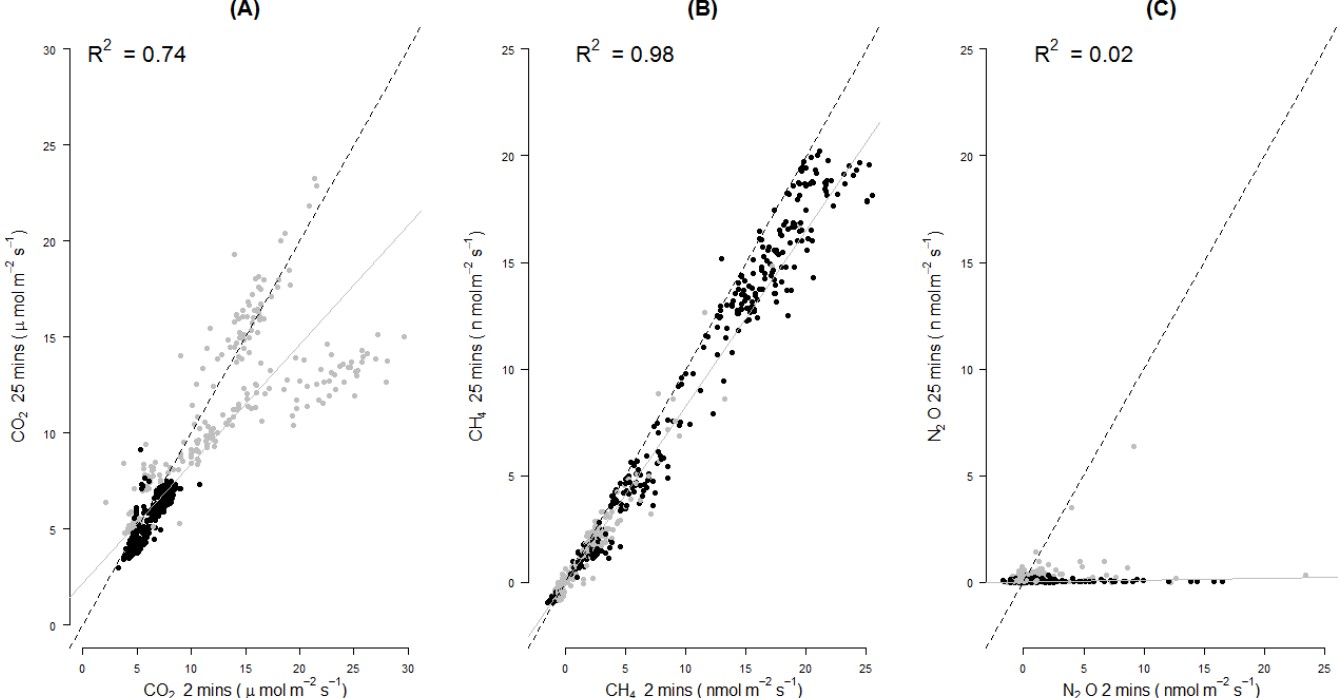

**Figure 2: Comparison of 2 minutes and 25 minutes estimations** for (A) $CO_2$ (B) $CH_4$ and (C) $N_2O$ fluxes. For this, we used measurements made over 25 minutes and recomputed the flux with the two firsts minutes for two weeks (from August 2[nd] for August 9[th] in black and from August 16[th] for August 25[th] in grey) covering the whole range of fluxes during the study period. All fluxes were computed using exponential fit. The dashed line represent the 1:1 line while the solid grey line represents the linear regression between 2 minutes and 25 minutes estimations ($R^2$ of these regressions are indicated on each panel).

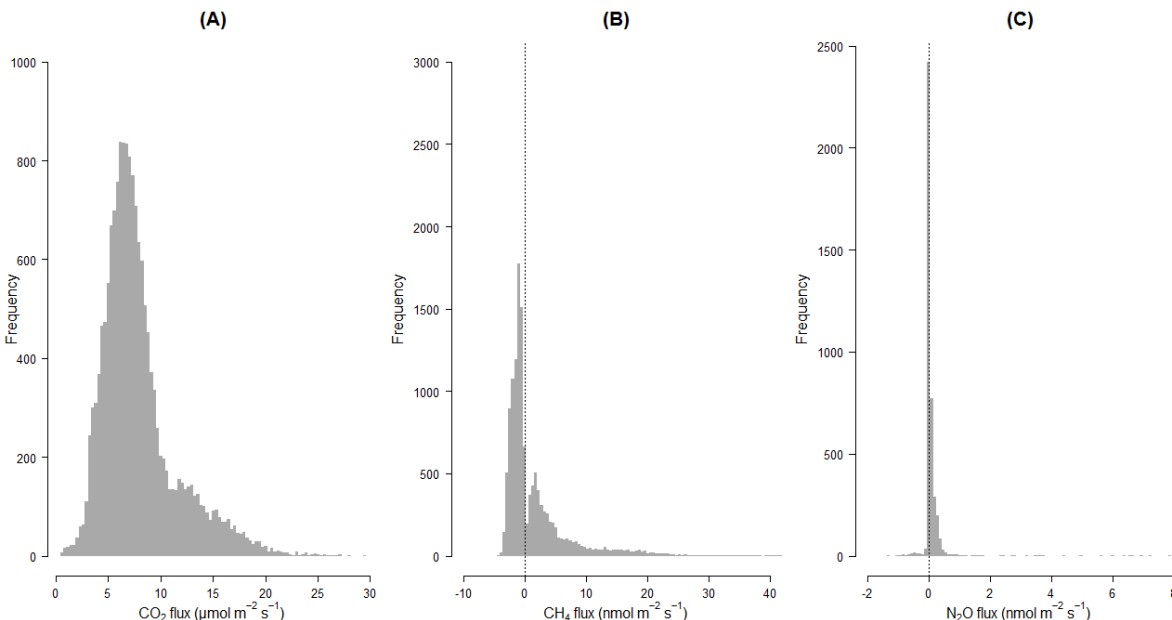

**Figure 3: Distribution of fluxes:** Histogram of (A) $CO_2$, (B) $CH_4$ and (C) $N_2O$ fluxes over the study period. For (B) and (C), the dotted line represents null fluxes.

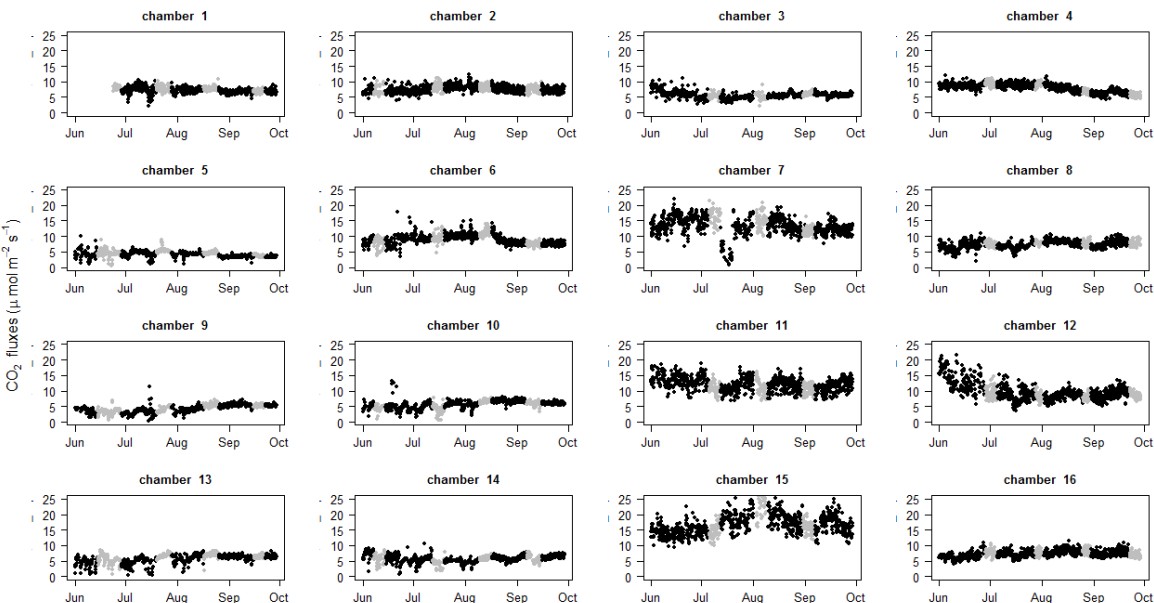

**Figure 4: CO₂ fluxes through time:** $CO_2$ fluxes for each chamber (1 to 16) over the study period with fluxes estimated with SHORT (2 minutes) closure time in black and fluxes estimated with the 2 first minutes of the LONG (25 minutes) closure time in grey. All panels have the same limits on the y axis (from 0 to 25 $\mu mol\ m^{-2}\ s^{-1}$)

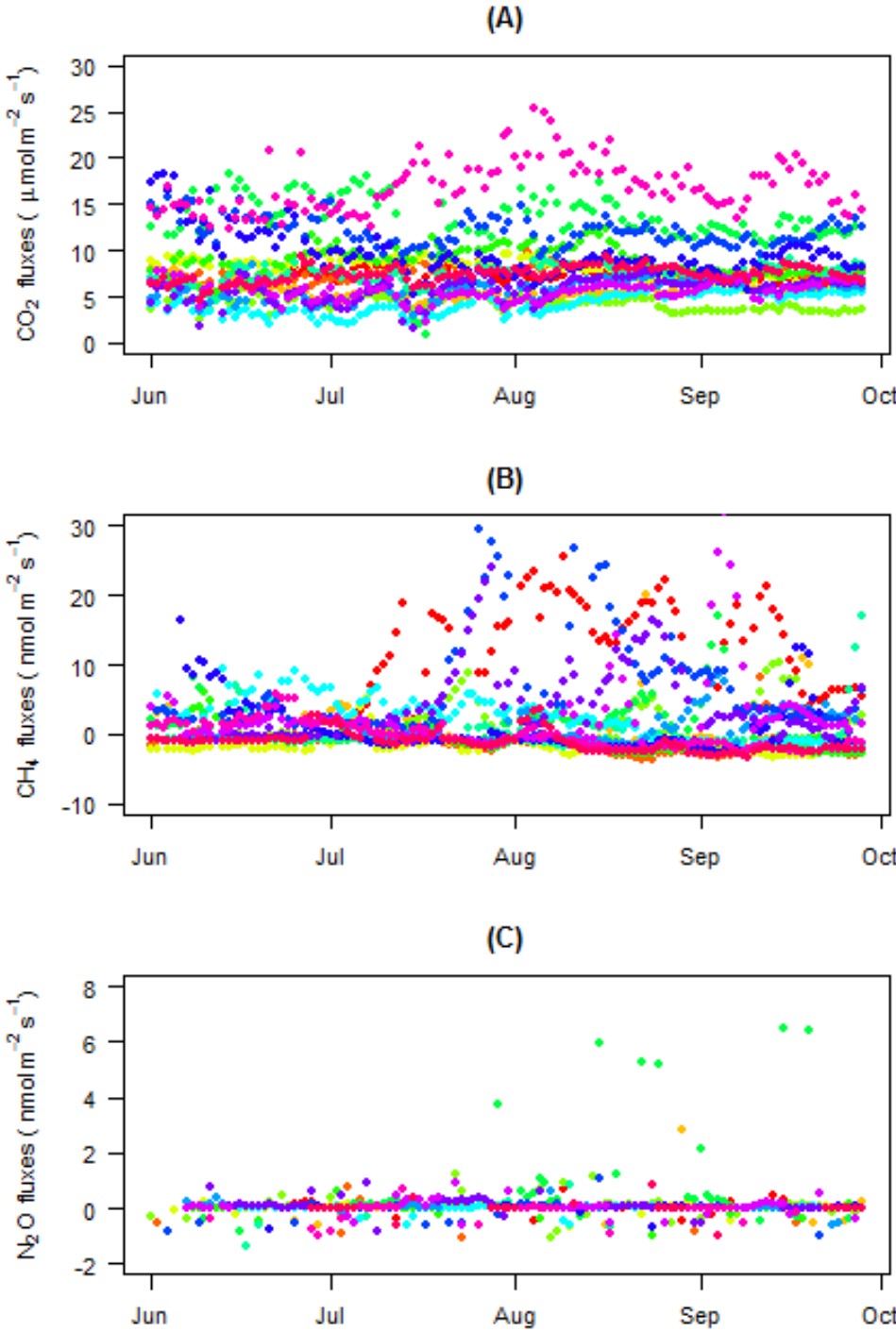

**Figure 5:** Mean values per days for (A) $CO_2$, (B) $CH_4$ and (C) $N_2O$ fluxes over the study period. Each chamber is represented by a distinct color.

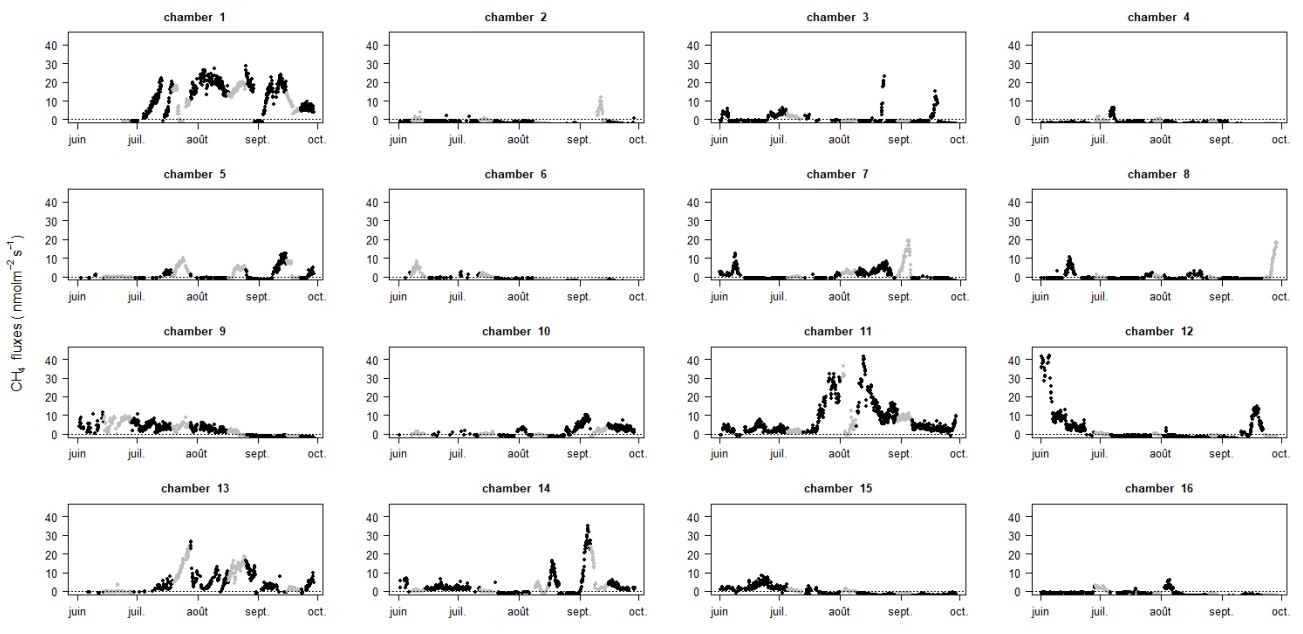

**Figure 6: CH₄ fluxes through time:** CH$_4$ fluxes for each chamber (1 to 16) over the study period with fluxes estimated with SHORT (2 minutes) closure time in black and fluxes estimated with LONG (25 minutes) closure time in grey. The dotted line displays the zero flux line. All panels have the same limits on the y axis (from -5 to 30 nmol m$^{-2}$ s$^{-1}$)

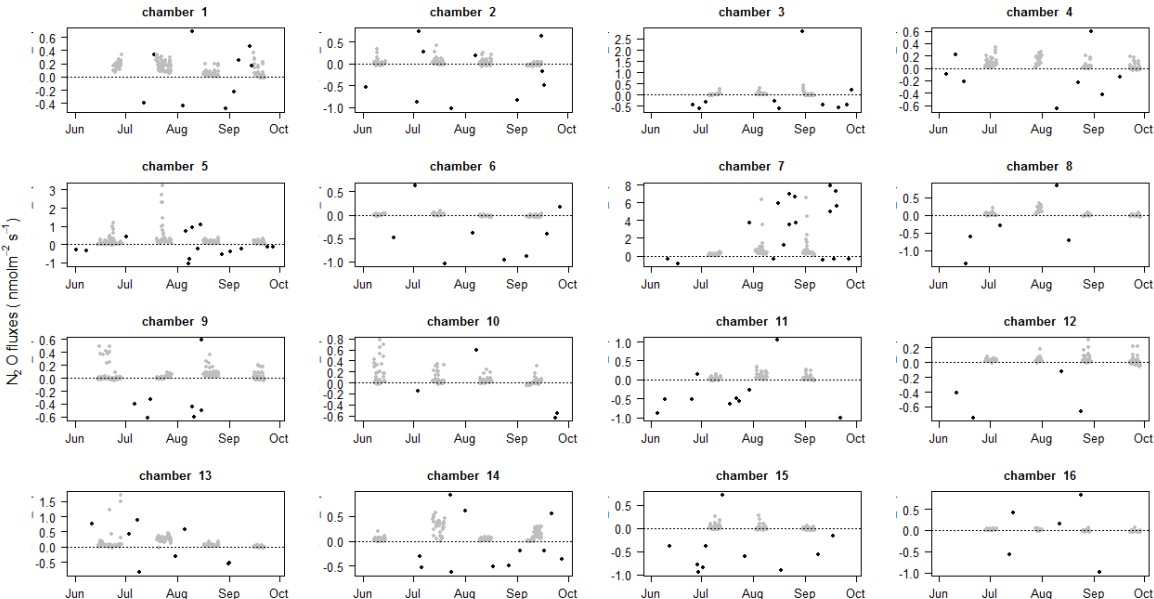

**Figure 7: N₂O fluxes through time:** N₂O fluxes for each chamber (1 to 16) over the study period with fluxes estimated with the SHORT (2 minutes) closure time in black and fluxes estimated with the LONG (25 minutes) closure time in grey. The dotted line displays the zero flux line. Due to the high differences among chambers, each panel has specific limit on the y axis.

**Supplementary File 1:** R code for merging Picarro files to include them in Soil Flux pro

```
## to list all the days in a given directory (Picarro makes one directory per day)
ListDay<-list.files()
Pfile<-list()
## to concatenate all the hourly file in one file per day
for (j in 1:length(ListDay))
{
print(j)
ListFilesPicarro<-list.files(ListDay[j])
Data<-read.table(paste(ListDay[j],"/",ListFilesPicarro[1],sep=""))
for (i in 2:length(ListFilesPicarro))
{
temp<-read.table(paste(ListDay[j],"/",ListFilesPicarro[i],sep=""))
Data<-rbind(Data, temp)
print(i)
}
Pfile[[j]]<-Data
}
## to concatenante all days and make just one file will all data
MasterData<-Pfile[[1]]
for (k in 2:length(Pfile))
{
MasterData<-rbind(MasterData,Pfile[[k]])
print(k)
}
## to write the table in a way that SFP can read it
write.table(MasterData, "MasterData.dat", quote=F)
```

**Supplementary Figure 1:** Comparison of linear (x-axis) and exponential (y-axis) fit of the same measurement for all the fluxes used in the study for (A) $CO_2$, (B) $CH_4$ and (C) $N_2O$. The dashed line represents the 1:1 line. High fluxes of all three gases are clearly underestimated using linear fit.

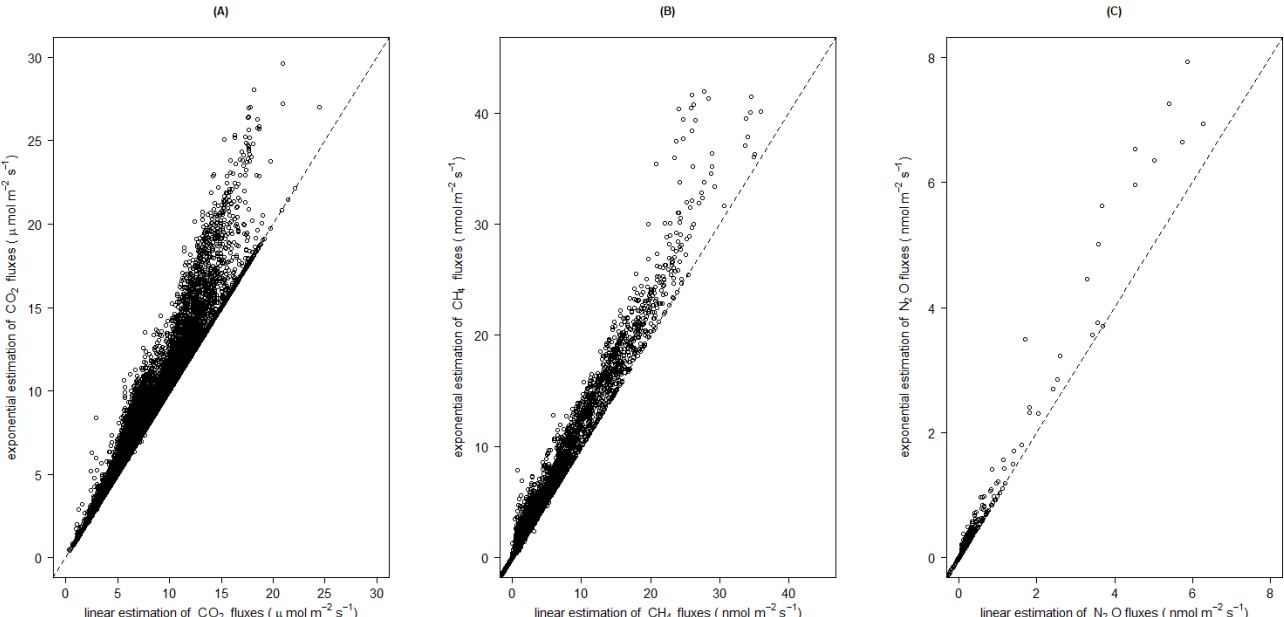

**Supplementary Figure 21:** Relationship between soil surface humidity and (A) $CO_2$, (B) $CH_4$ and (C) $N_2O$ fluxes over the study period.

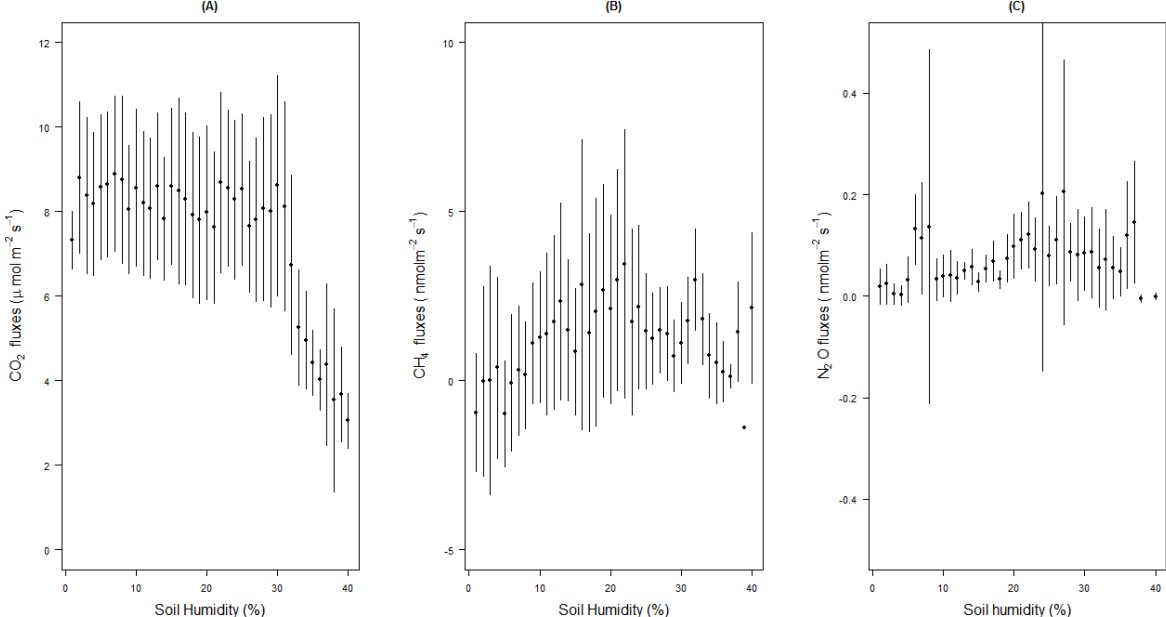

