# Peer review of "Automatic high-frequency measurements of full soil greenhouse gas fluxes in a tropical forest"

_Biogeosciences, 2018_

## Short Comment (SC1) · 27 Aug 2018

In the recent years, several studies highlighted the need for continuous measurements of soil GHG other than CO2, which has been technically challenging for long time. However, combination of different new instrumentation allows addressing this challenge nowadays. I think this manuscript a timely technical note addressing one of the most important issues regarding continuous measurements: which is the balance between frequency and reliability of measurements? Despite some of the points discussed in the paper are instrument specific considerations (Li8100 and G2308), I think that most of them apply for high-frequency studies using other instrumentation. In my

opinion, two points could be covered more in depth in order to make the manuscript more strong and inspiring for the community: (i) suitability of linear or exponential fits for estimating GHG fluxes, especially under high emissions and long chamber closure time; (ii) which threshold criterions do we have to apply for low rate fluxes and which are the consequences of using different criterions on temporal patterns (both short and long term scales) and on accumulated emission estimates. Finally, I want to recognize the challenge of running this complex instrument setup in a tropical forest. Dealing with high moisture when using IRGAS and CRDS is not easy, but the authors succeeded. I am looking forward to see the data in the full experiment context with their ecological implications. Here you could find some specific comments, suggestions and open discussion points: Pg3 L19-28. Li-8100 can detect really small fluxes of CO2 as well. I guess that the main reason for using both Li-8100 and Picarro G2308 is that one instrument controls the chambers and the other measures the three gases. Additionally, measuring simultaneously CO2 with two independent systems is a good control to validate the proper performance of the instruments. I wonder which was the agreement in CO2 between Li8100 and G2308. P4 L24-28. I don't know if I understand this statement, but SoilFluxPro (Li-COR software) allows to directly upload hundreds of Picarro files simultaneously (up to 2 months). You can choose to open all the files in one single file, and directly merge it with the Li-COR data. P4 L28-30. One of the best things of using SoilFluxPro is that calculates the fluxes using both linear and exponential fits, which could result in substantial differences in fluxes (see the attached example from my own data). My experience is that exponential equations usually fits better than linear ones (in terms of R2), especially for high flux rates under long chamber closure times.

P4 L29. Which was the actual length of each measurement without including the deadband? P5 L5. Why are you not using CO2 measured with Picarro? P5 L7-8. I guess that it has to be the volume of the system (chamber, Li8100, Picarro, multiplexer and tubing). This is really important since the volume of the system is a parameter controlling the minimum detectable flux, so Table 1 might substantially change depending

on this "detail". P5 L16-17. Again, this can be solved using exponential fits. P6 L7, L13 and L17. I guess these are not the correct figures. P6 L14-20. As far as I understood, you kept values higher than MDF (for emissions) and lower than –MDF (for sinks), but what happen with values close in between (-MDF < x < MDF)? What did you do with values close to 0 flux? And what happen if a flux was higher than MDF but had low R2? The same applies for N2O. Which criterion we have to use when measuring gas emissions at low rates? Is it a 0 flux, NA, should we keep the calculated flux regardless of the R2? Choosing one or other criterion might have several implications in order to estimate cumulative or mean fluxes, especially if the data does not have normal distribution and it's not 0 centered. In L11 you describe an R2 criterion for considering stable micrometeorological and chamber conditions based on CO2. Then, why we should apply other criterion for the other gases if the conditions are stable? I understand that for this might not be super relevant for a technical note, but this is a key question if you want to quantify emissions in natural conditions. In my opinion, this is the core of the study, and one of the most challenging issues we need to address when measuring CH4, N2O and other trace gases. When we have high fluxes, everything is clear, but when we have low fluxes, it turns more complicated. We were discussing this issue in Petrakis et al. 2017, but I still don't have the answers. I guess there is not a silver bullet. Figure 1A. There is something in this panel it's not completely clear. As far as I understand, the air goes from the chamber to the multiplexer, to Li8100, to G2308, to the external pump, to the multiplexer and again into the chamber. However, in the schematic view there is a black circuit (T piece sub-sampling loop) that connects the multiplexer, Li8100, G2308 and the external pump. Since the air composition does not change between these four elements, why the subsampling tub was not inserted in serial at one point of the circuit? Table 2. I wonder which closure time did you use in this table (2 or 25min). It would be interesting a comparison between 2 and 25 closure times. I'm not sure you will find differences in the means. This would suggest that short closure times might not affect the annual balance but deviation of the data (as we can see in Ap Figure 2). Appendix Figure A1. In my opinion, this is one of the most interesting figures in the manuscript and I think it should be place in the main manuscript. Some suggestions: a) Regressions will have better fit if you use exponential equations for estimating the flux. For each flux you can choose linear or exponential depending on the R2 b) Could you display R2 and the coefficients of the regressions between 2 and 25min? Regression B shows a good fit, but it seems that 2 min fluxes tends to overestimate fluxes compared to 25min estimates. Again, this could be an artifact of using linear regressions and not exponential. c) It would be interesting plotting the regression for N2O including all values (without removing data using R2 or MDF criterions)? This is related to my comment on Table 2. Apendix Figure A2. Please, edit the figure caption. Petrakis, S., Barba, J., Bond-Lamberty, B. and Vargas, R.: Using greenhouse gas fluxes to define soil functional types, Plant Soil, 1–10, 2017.

[Figure]

**Fig. 1.**

---

## Referee Comment (RC1) · Anonymous Referee #1 · 4 Sep 2018

This manuscript presents a protocol for measuring three greenhouse gases (CO2, CH4 and N2O) at high temporal frequency in a tropical forest soil using a combination of commercially available systems. This is a very timely and relevant manuscript, particularly for measurements of CH4 and N2O. Continuous high frequency measurements with clear sampling protocols will help researchers capture and model, transient changes in CH4 and N2O fluxes often missed by more infrequent sampling strategies. This looks like a nice, efficient system for measuring these important greenhouse gases. The technical write-up is comprehensive and easy to follow however I do have technical questions regarding their methodology that should be addressed. 1. How did the authors keep moisture from affecting sampling, either within the tubing lines

or when moving from IRGA to Picarro? Was there any moisture buildup in the tubing lines? 2. Were the instruments kept within operating temperature ranges: specifically the upper end of operating ranges? Where there diel changes in Licor and Picarro instrument temperatures? 3. How often were the instruments calibrated? 4. The authors subsampled from the flow downstream of the Li-8100A-IRGA into the Picarro G2308. What was the flow rate through the Picarro analyzer- was it 2.8L/min or some lower rate? I don't see the flow controller in Figure 1a diagram- is it built into the external pump? 5. What is the sampling volume inside the Picarro G2308? My concern is that if there is subsampling from a high flow rate at the T piece subsampling loop (Figure 1 a) through a secondary instrument, and then flow is re-merged downstream of the external pump and returned to the closed chamber there may be a dilution effect. This might not impact a 2 minute sampling time period but may have a greater impact on a 25minute time period. 6. Did the authors also use their Picarro G2308 to measure CO2 as well and if so how does that compare to the LI-8150 analyzer? This would provide confidence in running the two systems inline. Specific questions: Pg 2 line 26: are the reference [17,18] in the correct format? This is the only location that lists reference numbers as opposed to first author. Pg 3 line 23: the authors used an external pump, however the LI-8150 has an internal diaphragm pump- was this turned off or was it used inline with the external pump? Pg 3 line 28: the CRDS is not the "only" method that can detect low concentrations of N2O. Can you change to "one of the only" as opposed to "only". Pg 4 line 14: the flow rate of 2.8L/min is very high. Was this flow rate tested to determine if pressure within the closed chambers was altered? I assume that the Licor 8100 chamber tops had their patented pressure relief value installed? Figure 1b) are all the tubing lines from the 16 chambers to the multiplexer 15m (as in the text) in length? A small point but they look like different lengths in the diagram.

---

## Referee Comment (RC2) · Anonymous Referee #2 · 14 Sep 2018

**Review of the manuscript „Automatic high-frequency measurements of full soil greenhouse gas fluxes in a tropical forest"**

The authors measured *in situ* soil fluxes of $CO_2$, $CH_4$ and $N_2O$ continuously with a commercially available automated chamber system coupled with a CRDS analyser in a tropical forest for four months. The manuscript is focused entirely on the methodological aspect of these measurements, stressing the importance of adjusting chamber closure times for the different gases for reliable flux calculations. The effect of closure times on flux calculation results were studied by trying different chamber closure times in the field and by adjusting the number of data points used for the actual flux calculation.

This is a well-designed study, and overall, the manuscript is well written and structured. Also choosing appropriate chamber closure times and the operation of automated chamber systems are important topics for the soil flux community. However, I recommend publication of this manuscript after major revision because I have some general concerns with this manuscript.

General comments:
- Soil Flux Pro provides for each chamber measurement linear and non-linear flux calculations. Why did you choose to use only the linear flux calculation results? The underestimation of fluxes with linear regression due to saturation effects is well-known. That's why numerous non-linear calculation schemes have been developed. Could you have significantly reduced the chamber closure time for the $N_2O$ flux calculation if you had used non-linear flux calculation? The selection of the flux calculation scheme can change the MDF at least for chamber measurements with only few gas samples over time. Does this effect disappear with high-frequency analysers, i.e. selection of the flux calculation scheme becomes less crucial in that regard? Would there still be a significant difference between the SHORT and LONG flux calculation for the different gases when using non-linear flux estimates?
- You write about high-frequency measurements only as sampling measurement plots more frequently over time. However, you could also address the high-frequency sampling during a chamber closure since you use high-frequency gas analysers and work with MDF in your data analysis. There are several automated chamber systems which do not employ high-frequency analysers, but still collect discrete gas samples which have to be analysed with a GC. Especially for $N_2O$ it is very interesting to see what fluxes we can capture with CRDS in comparison to GC analysis.
- You only write how the SHORT and LONG measurements affected the flux estimates. But how did they affect the uncertainty of the single flux estimates? How large/small were the error bars for the flux estimates?
- Could you have just used one, namely the LONG, closure time for all chambers and only choose for the flux calculation between SHORT and LONG calculation times? This would be more practical than rotating closure times between chambers.
- In section 3, the results are clearly presented, but the discussion part is very limited.

Specific comments:
- Page 2, line 26: numbers instead of author names for references
- You are not always consisted in how you write company names (capital versus small letters). Also often you write 'minute' when you could just use 'min'.

– Page 3, line 23: What are the pump specifications? Was it the pump supplied by Picarro with the instrument or did you use another pump?

– Page 4, lines 9ff.: The soil temperature and soil moisture probes, were those the ones which can be directly attached to the chambers?

– Page 5, line 5ff.: Did you use the analytical accuracy specified on the technical datasheets of the analysers or did you perform measurements yourself? Where there significant air pressure and ambient temperature changes at your site over the four months? If yes, did you test how different temperature and pressure values could change the MDF estimate? Is $n$ incl. or excl. the deadband?

– Page 6, line 6: What $CO_2$ concentrations were reached during LONG closure times (and for $CH_4$)?

– Page 6, line 7: I find that confusing in comparison to section 2.5. So considering the deadband, the chambers were closed for 3 and 26 minutes, respectively?

– Page 6, line 20: Why did you consider these fluxes as unreliable when the chamber quality check using the $R^2$ for $CO_2$ was passed? Are you not unnecessarily filtering out fluxes which are not significantly different from zero, and thus introducing a bias in your data? Because this often happens when using $R^2$ as a filter criterion for low fluxes.

– Page 6, first paragraph of section 3: You had no problems with humidity and the automated chamber system at your site?

– Page 7, line 3: The conclusion about the 2 min sampling time sounds absolute, but it is only valid for your small chambers. Except for the necessary descriptions in the method section, you completely disregard the role of chamber volume for choosing the right chamber closure time.

– Page 7, lines 20/21: That sentence does not make any sense to me. 85.6 % of the fluxes were above or below?

– Page 8, line 4: You didn't show the diurnal variation in your data. This is more a point for the discussion than a conclusion from your presented data.

– The references are not well formatted.

– Table 1: Use superscript for the units.

– Table 2: Include $n$ for each chamber.

– Check how the units are written on the y-axis of the figures.

---

## Referee Comment (RC3) · Anonymous Referee #3 · 3 Oct 2018

In this manuscript, the authors detail a field-deployed and field-tested system for measuring soil greenhouse gas (GHG) emissions (CO2, CH4 and N2O) from a tropical wet forest; the system leverages a commercially-available automated flux chamber system with a commercially-available CRDS analyzer. More specifically, the authors (a) outline the technical protocol for implementing such a system, (b) report the mean fluxes and variability of CO2, CH4 and N2O observed over the four-month deployment period and (c) test two chamber closure lengths to determine the most effective experimental design for capturing fluxes above minimum detectable levels. Successfully implementing such a system in the tropics is both difficult and has only been done rarely, so a technical note detailing how to do so is absolutely a contribution to the literature.

[Figure]

I have two general/more broad comments about the manuscript, which I detail below, and also include several more specific comments at the end of this review.

1. CONCERNS REGARDING FLUX CALCULATION PROTOCOLS

In these automated, high-frequency GHG systems, one important set of experimental design and protocol decisions govern how to calculate flux rates and screen for acceptable data points. The authors lay out fairly transparent information about how they calculated their flux rates for each gas, but I wonder if more discussion of the implications of their calculation choices is warranted. I have a few specific questions. Am I correct in understanding that the authors only calculated flux rates for all three gases based on a linear model (Pg 4, Ln 27)? If so, I wonder why they didn't consider also fitting exponential models to the CH4 and N2O fluxes, if not the CO2 fluxes. The authors themselves note on Pg 5, Ln 17 that using certain chamber closure times (which, of course, this paper is very interested in) in combination with a linear flux fit can lead to flux underestimation. Couldn't the "optimal" chamber closure time that the authors attempt to find also include some experimental designs with different closure lengths but non-linear flux fits?

Additionally, the Picarro G2308 records numerous diagnostic variables alongside GHG concentrations, including measures of moisture, temperature, pressure, etc. From methods section 2.6, I am under the impression that fluxes were only struck from the dataset if they were (a) below the MDF, (b) had an R2 for CO2 < 0.9, or (c) only struck N2O data if the SHORT R2 < 0.8. First, the authors might consider including a supplemental figure that justifies their decision not to have data quality rules around CH4, as they do for N2O. Second, like the other reviewers, I was curious as to how humidity was dealt with, since it appears that the moisture-related diagnostics weren't used to evaluate data quality. Was there a water trap that isn't marked in the instrument set-up diagram?

More broadly, as this paper aims to outline best practices for setting up this kind of

experiment in the tropics, fleshing out the data management aspect of things would improve the paper, in my opinion.

2. TECHNICAL NOTE VS. DATA EXPLORATION PAPER

The aspects of this paper that serve as a technical note are novel and helpful. That said, the results and discussion section, in which the observed GHG fluxes are analyzed, is perfunctory and is relatively focused on a methodological question: what chamber closure time should be used in this system, and how can others determine what chamber closure time to use in their analogous system? I found myself wishing that there was a more robust analysis of the GHG data itself and the ecological implications of their various findings. See also my comment below about Table 2.

SPECIFIC COMMENTS

* Pg 7, Ln 30: The authors ultimately recommend a sampling protocol that rotates between short and long closure times. What is preventing them from recommending always doing LONG chamber closures and only using the first two minutes of chamber closure time to calculate the $CO_2$ flux, thus decreasing the amount of human labor needed to swap out the system program once a week?

* Pg 7, Ln 23 / Figure A2: This figure is used to justify why LONG chamber readings weren't reliable for $N_2O$ flux estimation, but these data don't indicate that the variable fluxes are unreliable, only that they are variable and considerably larger in magnitude than the SHORT $N_2O$ flux estimates. Can an additional supplemental figure be added showing the $R^2$ values for the LONG vs. SHORT $N_2O$ fluxes? Or some similar figure that shows why the LONG fluxes are considered unacceptable?

* Pg 7, Ln 31: "Our unique system..." and Pg 8, Ln 12: "this is the first time that this experimental set up is described and tested under tropical field conditions." I believe an analogous system was described in Puerto Rico (O'Connell et al 2018, Nature Communications, https://doi.org/10.1038/s41467-018-03352-3), though not as a technical

or methods paper.

* Appendix Figure A1: Might be worth including the N2O comparison just as the CO2 comparison is included even though the authors discard the LONG CO2 flux estimates.

* Table 2: A number of the authors' chambers reported mean N2O fluxes below 0. This seems worth mentioning in the results and/or discussion.

---

## Short Comment (SC2) · 29 Oct 2018

This manuscript focused on a very important topic about soil CO2/CH4/N2O fluxes in tropical rainforest. The experiment was well designed. Particularly, this may be the world's first report about in situ and simultaneously measurement of soil CO2/CH4/N2O fluxes at low latitude (between 10° N and 10° S). I would like to give the authors my comments.

1. Important references:

To date, through the "Web of Science", I could not find any publication about continuous

measurement of soil $CO_2$ efflux (Rs) using the automated chambers in the low latitude tropical forests that between 10° N and 10° S. Though two campaign studies in very humid forests (ïĆş3500 mm of annual precipitation) using automated chambers each in northeastern Australia (17° S) (Kiese and ButterbachBahl, 2002) and northeastern Puerto Rico (18° N) (Wood et al., 2013) were conducted only less than 6-month period, they observed similar phenomenon with Rs was higher during the dry season but lower during the wet season. Kiese and ButterbachBahl (2002) also measured $N_2O$ flux. Conversely, a 4-year continuous measurement of Rs in a seasonal dry (1,250 mm of annual precipitation) tropical forest in western Thailand (14° N) showed higher Rs in wet season than that of dry season (Hanpattanakit et al., 2015).

2. $CO_2$ flux:

Empirically, also see the above references, $CO_2$ flux is largely controlled by soil moisture (rain events) at tropical forests. However, based on Fig 3, during 4-month experiment (June-September 2016), most of the chambers did not show temporal variation in $CO_2$ flux. Thus, the authors are suggested to add soil moisture (and temperature) data to Fig 3 and provide some discussion about the (lack of) relationships between Rs and soil moisture and temperature.

3. $CH_4$ flux:

Generally speaking, upland forest soil is a $CH_4$ sink, even lowland tropical forest soil. Compared to Rs, however, $CH_4$ flux is more complex and generally has large spatial variation, because the termite activity can emit $CH_4$ thus offset a partial of the soil $CH_4$ sink. I am confused with Table 2, because ten of the sixteen chambers showed $CH_4$ source. Li-Cor soil chamber (8100-104) can be considered to block most activity of the termite, because the chamber base (collar; 7 cm in height) was inserted ∼7 cm into the soil and left another 4 cm above the soil; in addition, the chamber has relative additional big metal base surround the collar. On the other hand, inserted chamber base (collar) into the tropical (clay) soil can (sometimes) cause waterlogging inside the Li-Cor soil

chamber (8100-104), which might convert the CH4 sink to CH4 source. Same with CO2 flux, temporal variations in CH4 fluxes also could not be detected in Fig. 4. Also, megascopically, the chambers did not show the common pattern of temporal variation in CH4 fluxes (Fig 4). Sure, this forest has plentiful precipitation (about 3000 mm) and very low elevation, both of these abiotic factors may cause the site as CH4 source. Thus, the authors are suggested to provide some more discussion about (the lack of) spatio-temporal variation in CH4 flux.

4. Appendix Figure A1:

This figure shows a very general (basic) chamber-problem for measurement of soil GHGs fluxes. Long closure time will cause higher GHGs concentration (if the soil is GHGs source) or lower GHGs concentration (if the soil is GHGs sink) inside the chamber, which will induce underestimation of GHGs flux (saturation effect). Saturation effect is generally positively associated with both flux rate and ratio of the effective chamber volume to the measured soil surface area. Empirically, I believe the 2-mintute closure time is enough for measurement of both CO2 and CH4 flux in tropical forests, even for most temperate and boreal forests. For Li-Cor soil chamber (8100-104), the ratio is (0.0040761/ 0.03178=0.12826 m) = 12.3 cm. However, for many of the custom-made soil chambers, the ratio is generally higher than 12.3 cm, thus this is might be the specific problem (issue) only for Li-Cor soil chamber (8100-104). I suggest the authors feedback this problem to Li-Cor and suggest Li-Cor to draw this problem to their instrument user manual.

5. Also for Appendix Figure A1:

The authors are suggested to re-draw the Appendix Figure A1 indicating different symbols (or color) for each of the four chambers.

6. Closure time:

When compared Table 1 with Table 2, the closure time of 10 minutes for measurement

of N2O flux was enough. Thus, the Table 1 is suggested to be deleted.

7. Additional suggestion 1:

To prove the data quality or measurement precision, the authors are suggested to add a plot showing changes in CO2, CH4 and N2O concentrations in the chambers. Following is a sample plot (Sample Fig).

8. Additional suggestion 2:

As I mentioned in the above, this may be the world's first report about in situ and simultaneously measurement of soil CO2/CH4/N2O fluxes at low latitude (between 10° N and 10° S). I believe this paper will be a potential high citation rate if the authors can give some more discussion about spatio-temporal variation in CO2/CH4/N2O fluxes and their control factors. For example, the coefficient of variation (CV) was used to represent the spatial variation. CV of Rs can be calculated by CV = (SD/(mean Rs)) ×100.

9. Useful reference:

(1) Hanpattanakit, P. et al., 2015. Multiple timescale variations and controls of soil respiration in a tropical dry dipterocarp forest, western Thailand. Plant and Soil, 390(1-2): 167-181.

(2) Kiese, R. and ButterbachBahl, K., 2002. N2O and CO2 emissions from three different tropical forest sites in the wet tropics of Queensland, Australia. Soil Biology & Biochemistry, 34(7): 975-987.

(3) Wood, T.E., Detto, M. and Silver, W.L., 2013. Sensitivity of Soil Respiration to Variability in Soil Moisture and Temperature in a Humid Tropical Forest. PLoS One, 8(12): 7.
* * *
[Figure]

**Sample Fig.** Changes in $CO_2$ and $CH_4$ concentration inside the chambers. This is an *in situ* soil $CO_2$ and $CH_4$ flux measurement in a low latitude (<3° N) lowland rainforest using a 16-channel custom-made automated chamber (70×70×50 cm, L×W×H) system coupled with an IRGA $CO_2$ analyzer (LI-820, Li-Cor Biosciences) and a cavity ring down spectroscopy $CO_2/CH_4/H_2O$ analyzer (UGGA, LGR). Half (eight) of the sixteen chambers were trenched for measurement of heterotrophic respiration. Figure shows one measurement cycle (1 hour) for the sixteen chambers with each sequentially closed for 225 s. $CO_2$ concentration measured by both LI-820 and UGGA shows linearly increased for all chamber during the 225 s closure time (a). $CH_4$ concentration (b) shows linearly decreased in all trenched and half (four) of the control chambers but increased in another half of the control chambers (red circles).

**Fig. 1.** Sample Fig

---

## Author Comment (AC1) · 9 Nov 2018

This manuscript presents a protocol for measuring three greenhouse gases ($CO_2$, $CH_4$ and $N_2O$) at high temporal frequency in a tropical forest soil using a combination of commercially available systems. This is a very timely and relevant manuscript, particularly for measurements of $CH_4$ and $N_2O$. Continuous high frequency measurements with clear sampling protocols will help researchers capture and model, transient changes in $CH_4$ and $N_2O$ fluxes often missed by more infrequent sampling strategies. This looks like a nice, efficient system for measuring these important greenhouse gases.

[Figure]

Response: We thank the reviewer for the positive comments and constructive inputs.

The technical write-up is comprehensive and easy to follow however I do have technical questions regarding their methodology that should be addressed.

1. How did the authors keep moisture from affecting sampling, either within the tubing lines or when moving from IRGA to Picarro? Was there any moisture buildup in the tubing lines?

Response: We also worried about moisture problems prior to our experiment, but such problems never occurred. No significant condensation occurred in our system, we assume because temperature variations were typically small below the canopy, varying from 22°C in the night to 28°C during the day (see Figure attached). We regularly checked that no liquid water accumulation occurred in the sampling tubes. Moreover, both analysers measure water vapor and account for its effect on concentration of $CO_2$, $CH_4$ and $N_2O$ .

Changes in the manuscript: Temperature variations are typically small below the canopy due to the shadowing by dense canopy crown and microclimatic conditions. During the study period, temperature at 2m height varies during the day from 22 °C in the night to 28 °C during the day. The presence of water condensation inside the tubing lines was carefully checked every week and never occurred during the study period.

2. Were the instruments kept within operating temperature ranges: specifically the upper end of operating ranges? Where there diel changes in Licor and Picarro instrument temperatures?

Response: In our study site, all the systems were operating below dense understory vegetation and canopy cover, which naturally create a buffer maintaining air temperature relatively constant over the year (i.e. at daily and seasonally time scale). There were no diel changes in Licor and Picarro inside operating temperature: Picarro temperature was monitored and remained between 44.99 °C and 45.01 °C and Licor temperature at 51.7 °C.

3. How often were the instruments calibrated?

Response: All the systems used at this time, i.e. from June to September 2016, were new and received from the manufacturers. We therefore did not re-calibrate them during the study period. We are confident that the data of the gas concentrations recorded by our analysers were robust.

4. The authors subsampled from the flow downstream of the Li-8100A-IRGA into the Picarro G2308.What was the flow rate through the Picarro analyzer- was it 2.8L/min or some lowerrate? I don't see the flow controller in Figure 1a diagram- is it built into the external pump?

Response: The flow rate of 2.8L/min corresponds to the flow between the chambers and the multiplexer which cannot be adjusted. This high rate inside the chamber allows to achieve a sufficient air mixing in the chamber headspace during the measurements. Flow rates in the subsampling lines (Li8100 and Picarro) were lower and set between 1.5 and 1.7L/min as recommended by the manufacturers.

Changes in the manuscript: Flow rates in the subsampling lines (Li8100 and Picarro) were lower and set between 1.5 and 1.7 L min-1 as recommended by the companies.

5. What is the sampling volume inside the Picarro G2308? My concern is that if there is subsampling from a high flow rate at the T piece subsampling loop (Figure1 a) through a secondary instrument, and then flow is re-merged downstream of the external pump and returned to the closed chamber there may be a dilution effect. This might not impact a 2 minute sampling time period but may have a greater impact on a 25 minute time period.

Response: The volume of the PICARRO analyser together with tubing from the subsampling loop represents 130 cm3. It represents only 2% of the total volume (6088 cm3) and the dilution effect was therefore limited.

6. Did the authors also use their Picarro G2308 to measure CO2 as well and if so how does that compare to the LI-8150 analyzer? This would provide confidence in running the two systems inline.

Response: One step of our data quality check / quality control consisted in comparing the soil CO2 effluxes measured by the PICARRO with the soil CO2 effluxes measured by the Li-8100A. This method was also used to control that there were no leaks. Overall, both systems agreed very well. However, while the PICARRO G2308 analyser automatically reports dry mol fraction of CH4 and N2O, it only reports CO2 uncorrected by H2O concentration. As the precision of measurement was better for CO2 using the Li8100 and as it also automatically reports dry mol fraction for this gas, we decided to use Picarro estimation for CH4 and N2O and Li8100 estimation for CO2.

Specific questions: Pg 2 line 26: are the reference [17,18] in the correct format? This is the only location that lists reference numbers as opposed to first author.

Response: This has been corrected

Pg 3 line 23: the authors used an external pump, however the LI-8150 has an internal diaphragm pump- was this turned off or was it used in line with the external pump?

Response: The system was always operating with the two pumps, the internal diaphragm pump of the Li-8150 and the external pump of the PICARRO, turned on, which limited the risk of water condensation inside the tubing line.

Pg 3 line 28: the CRDS is not the "only" method that can detect low concentrations of N2O. Can you change to "one of the only" as opposed to "only".

Response: This has been corrected.

Pg 4 line 14: the flow rate of 2.8L/min is very high. Was this flow rate tested to determine if pressure within the closed chambers was altered? I assume that the Licor 8100 chamber tops had their patented pressure relief value installed?

Response: The flow rate of 2.8L/min corresponds to the flow between the chambers and the multiplexer which cannot be adjusted. This high rate inside the chamber allows to achieve a sufficient air mixing in the chamber headspace during the measurements. Flow rates in the subsampling lines (Li8100 and Picarro) were lower and set between 1.5 and 1.7L/min as recommended by the manufacturers.

Figure1b) are all the tubing lines from the 16 chambers to the multiplexer 15m (as in the text)in length? A small point but they look like different lengths in the diagram.

Response: Yes the length of tubing was 15 m for all chambers but as chambers are installed in a grid around the instruments, they are not all 15m away from the instruments. The length of the tubing lines (15 m) is cited in the caption.

Please also note the supplement to this comment:
https://www.biogeosciences-discuss.net/bg-2018-341/bg-2018-341-AC1-supplement.pdf

―――――――――――――――

[Figure]

*Figure 1: Daily temperature variation at 2m height under the canopy cover during the study period (June-September 2016).*

**Fig. 1.**

---

## Author Comment (AC3) · 9 Nov 2018

The authors measured in situ soil fluxes of CO2, CH4 and N2O continuously with a commercially available automated chamber system coupled with a CRDS analyser in a tropical forest for four months. The manuscript is focused entirely on the method-ological aspect of these measurements, stressing the importance of adjusting chamber closure times for the different gases for reliable flux calculations. The effect of closure times on flux calculation results were studied by trying different chamber closure times in the field and by adjusting the number of data points used for the actual flux calcu-lation. This is a well-designed study, and overall, the manuscript is well written and

structured. Also choosing appropriate chamber closure times and the operation of automated chamber systems are important topics for the soil flux community. However, I recommend publication of this manuscript after major revision because I have some general concerns with this manuscript.

Response: We express our deep thanks to reviewer for his positive comments about our manuscript and constructive remarks. We have addressed, see below, our answers to each comment / remark.

1. Soil Flux Pro provides for each chamber measurement linear and non-linear flux calculations. Why did you choose to use only the linear flux calculation results?

Response: We firstly decided to use linear flux calculations only because we thought that the saturation effects characterised by a plateau after a certain time would be low. However, following your comment and comment from other reviewers, we changed flux calculations in the new version of the manuscript to use exponential estimations.

The underestimation of fluxes with linear regression due to saturation effects is well-known. That's why numerous non-linear calculation schemes have been developed. Could you have significantly reduced the chamber closure time for the N2O flux calculation if you had used non-linear flux calculation?

Response: See previous comment.

The selection of the flux calculation scheme can change the MDF at least for chamber measurements with only few gas samples over time. Does this effect disappear with high-frequency analysers, i.e. selection of the flux calculation scheme becomes less crucial in that regard? Would there still be a significant difference between the SHORT and LONG flux calculation for the different gases when using non-linear flux estimates?

Response: The standard error approach that we used (Nickerson, 2016) is a first order approximation for the MDF from high-frequency measurements and the "true" MDF is a function of the chamber time-series fit type as well (i.e. Linear, exponential, quadratic).

Nonetheless, while the use of linear regression resulted in systematically smaller fluxes as compared to exponential regression. It is therefore recommended to initially calculate fluxes with linear regression to determine the threshold for "low" fluxes and to recalculate them using exponential regression (Korkiakoski et al., 2017).

Korkiakoski, M., Tuovinen, J.-P., Aurela, M., Koskinen, M., Minkkinen, K., Ojanen, P., Penttilä, T., Rainne, J., Laurila, T. and Lohila, A.: Methane exchange at the peatland forest floor–automatic chamber system exposes the dynamics of small fluxes, 2017. Nickerson, N.: Evaluating gas emission measurements using Minimum Detectable Flux (MDF), Eosense Inc Dartm. N. S. Can., 2016. 2. You write about high-frequency measurements only as sampling measurement plots more frequently over time. However, you could also address the high-frequency sampling during a chamber closure since you use high-frequency gas analysers and work with MDF in your data analysis. There are several automated chamber systems which do not employ high frequency analysers, but still collect discrete gas samples which have to be analysed with a GC. Especially for N2O it is very interesting to see what fluxes we can capture with CRDS in comparison to GC analysis.

Response: In a previous study in the same environment (Courtois et al., 2018), we estimated that the minimum detectable fluxes using Gas Chromatography analysis of four discrete gas samples over 30 minutes for N2O was $\pm$ 8.3 $\mu$g N m$-2$ h$-1$. MDF estimated in the present study using high frequency measurement was 0.002 nmol m-2 s-1 or 0.2 $\mu$g N m-2 h-1 for N2O which is therefore $\sim$ 40 times lower. We added a sentence in the manuscript to highlight this interesting fact.

Courtois, E. A., Stahl, C., Van den Berge, J., Bréchet, L., Van Langenhove, L., Richter, A., Urbina, I., Soong, J. L., Peñuelas, J. and Janssens, I. A.: Spatial Variation of Soil CO2, CH4 and N2O Fluxes Across Topographical Positions in Tropical Forests of the Guiana Shield, Ecosystems, 1–14, 2018. 3. You only write how the SHORT and LONG measurements affected the flux estimates. But how did they affect the uncertainty of the single flux estimates? How large/small were the error bars for the flux estimates?

Response: Comparison of standard error of single flux estimates using 2 minutes or 25 minutes estimations for two weeks (from August 2nd for August 9th and from August 16th for August 25th) shows that standard errors are always higher for 2 minutes than for 25 minutes estimation for all three gases (Figure 2 below). Nonetheless, we decided not to integrate this figure in the manuscript because it does not add much to the study.

4. Could you have just used one, namely the LONG, closure time for all chambers and only choose for the flux calculation between SHORT and LONG calculation times? This would be more practical than rotating closure times between chambers.

Response: Setting all chambers as LONG measurements would have led to a maximum of ∼ 3 measurements only per chamber and per day. Mixing LONG and SHORT measurements allows to maximise the number of measurements per chamber and per days while ensuring a reliable estimation of the low N2O fluxes and to capture transient peaks of CH4 and N2O.

5. In section 3, the results are clearly presented, but the discussion part is very limited.

Response: The main aim of our study was not to identify controls and mechanisms of the soil GHG fluxes but rather to test novel soil GHG systems for continuous high-frequency measurements. We think that this manuscript could be used as technical support to set up new soil systems and contribute to record comparable soil GHG data in other regions around the world. Nonetheless, the discussion has been revised in the new version of the manuscript to integrate discussion on spatio-temporal variability of fluxes based on our study.

Specific comments: Page 2, line 26: numbers instead of author names for references

Response: This has been corrected.

You are not always consisted in how you write company names (capital versus small letters). Also often you write 'minute' when you could just use 'min'.

Response: We corrected this in the new version of the manuscript.

Page 3, line 23: What are the pump specifications? Was it the pump supplied by Picarro with the instrument or did you use another pump?

Response: We have included more information in the manuscript about the external pump provided by PICARRO: recirculation pump A0702

Page 4, lines 9.: The soil temperature and soil moisture probes, were those the ones which can be directly attached to the chambers?

Response: The soil temperature and soil moisture probes were those provided by Li-COR, which are directly attached to the chambers. The probes measured soil temperature and soil moisture around the PVC collars. We have added more details in the text.

Page 5, line 5.: Did you use the analytical accuracy specified on the technical data sheets of the analysers or did you perform measurements yourself?

Response: We used the analytical accuracy specified on the technical data sheets of the analysers. This is now specified in the new version of the manuscript.

Where there significant air pressure and ambient temperature changes at your site over the four months? If yes, did you test how different temperature and pressure values could change the MDF estimate? Is nincl. or excl. the deadband?

Response: Please, see our response to reviewer 1 above (2.). In our study site, because all the systems were operating under dense understory vegetation and canopy cover, air temperature remained relatively constant over the year (i.e. at daily and seasonally time scale) near the soil surface. This is also true for air pressure.

Page 6, line 6: What $CO_2$ concentrations were reached during LONG closure times (and for$CH_4$)?

Response: $CO_2$ concentration can reach 2000 ppm and $CH_4$ concentration 4 ppm.

Page 6, line 7: I find that confusing in comparison to section 2.5. So considering the

deadband, the chambers were closed for 3 and 26 minutes, respectively?

Response: No, the chamber stayed close for 2 minutes and 25 minutes and the first minute was not used for the flux estimation. As we have a sampling frequency of 1 second, it still represents 60 points for curve fitting. Nonetheless, we agree that this could be considered as a too short period for CH4 and CO2 estimation using the SHORT (2 minutes) closure time. We therefore compared the CO2 and CH4 estimations with a deadband of 60 seconds (fluxes estimation with 60 seconds) with a deadband of 30 seconds (fluxes estimation with 90 seconds) for the week from August 16th to August 25th. These two estimations were very well correlated (see figure 3 below) so we decided to keep our 60 s deadband results.

Page 6, line 20: Why did you consider these fluxes as unreliable when the chamber quality check using the $R^2$ for CO2 was passed? Are you not unnecessarily filtering out fluxes which are not significantly different from zero, and thus introducing a bias in your data? Because this often happens when using $R^2$ as a filter criterion for low fluxes.

Response: Because of the high soil respiration activity, low soil CO2 fluxes do not really occur in this tropical rainforest, not even during the dry season. When the R2 criterion for CO2 was not passed, it always corresponded to situations of imperfect closure of the chamber, due to leaves or small branches lying on the soil collars (381 measurements over 17796, i.e. 2.1%). In these cases, it was therefore necessary to remove flux data for the three gases.

Page 6, first paragraph of section 3: You had no problems with humidity and the automated chamber system at your site?

Response: Please, see our response to reviewer 1 above (1.); at the given flow rate and the small diel cooling, we had no problems with water condensation inside the tubing lines of our system.

Page 7, line 3: The conclusion about the 2 min sampling time sounds absolute, but

it is only valid for your small chambers. Except for the necessary descriptions in the method section, you completely disregard the role of chamber volume for choosing the right chamber closure time.

Response: We added a sentence to precise that our result is valid for small chambers only.

Page 7, lines 20/21: That sentence does not make any sense to me. 85.6 % of the fluxes were above or below?

Response: This sentence has been considerably modified.

Page 8, line 4: You didn't show the diurnal variation in your data. This is more a point for the discussion than a conclusion from your presented data.

Response: We agree with the reviewer; however, ecological interpretation of our data will require more long-term data and will be published in a future paper. Here, we wanted to provide a technical report with information on how to get robust results on GHG flux estimations rather than on how these fluxes are produced and vary.

The references are not well formatted.

Response: References were reformatted

Table 1: Use superscript for the units.

Response: This has been corrected

Table 2: Include n for each chamber.

Response: This information has been added.

Check how the units are written on the y-axis of the figures.

Response: This has been checked..
* * *
[Figure]

*Figure 2: Comparison of standard error of single flux estimates using 2 minutes or 25 minutes estimations for two weeks (from August 2nd for August 9th and from August 16th for August 25th)*

**Fig. 1.**

[Figure]

*Figure 3: Comparison of CO2 and CH4 fluxes with a 30 seconds and a 60 seconds deadband
for the week from August 16th to August 25$^{th}$. The red dotted lines represents the 1:1 line.*

**Fig. 2.**

---

## Author Comment (AC4) · 9 Nov 2018

In the recent years, several studies highlighted the need for continuous measurements of soil GHG other than CO2, which has been technically challenging for long time. However, combination of different new instrumentation allows addressing this challenge nowadays. I think this manuscript a timely technical note addressing one of the most important issues regarding continuous measurements: which is the balance between frequency and reliability of measurements? Despite some of the points discussed in the paper are instrument specific considerations (Li8100 and G2308), I think that most of them apply for high-frequency studies using other instrumentation. In my

opinion, two points could be covered more in depth in order to make the manuscript more strong and inspiring for the community:

(i) suitability of linear or exponential fits for estimating GHG fluxes, especially under high emissions and long chamber closure time

Response: We first decided to use linear flux calculations only because we thought that the saturation effects characterised by a plateau after a certain time would be low. However, following your comment and comment from other reviewers, we changed flux calculations in the new version of the manuscript to use exponential estimations.

(ii) which threshold criterions do we have to apply for low rate fluxes and which are the consequences of using different criterions on temporal patterns (both short and long term scales) and on accumulated emission estimates.

Response: Some information regarding this issue were added in the new version of the manuscript.

Finally, I want to recognize the challenge of running this complex instrument setup in a tropical forest. Dealing with high moisture when using IRGAS and CRDS is not easy, but the authors succeeded.

Response: Thank you for this positive comment.

I am looking forward to see the data in the full experiment context with their ecological implications. Here you could find some specific comments, suggestions and open discussion points: Pg3 L19-28. Li-8100 can detect really small fluxes of $CO_2$ as well. I guess that the main reason for using both Li-8100 and Picarro G2308 is that one instrument controls the chambers and the other measures the three gases. Additionally, measuring simultaneously $CO_2$ with two independent systems is a good control to validate the proper performance of the instruments. I wonder which was the agreement in $CO_2$ between Li8100 and G2308.

Response: In our system, the automated chambers were controlled by the Li-8150,

which was controlled by the Li-8100A. The gas analysers were Li-8100A for the CO2 and PICARRO G2308 for the CH4 and N2O. We recommend using the Li-8100A to determine the soil CO2 effluxes but recognise that CO2 information provided by the PICARRO can still be used to check for potential leaks in the analysers / tubing.

P4 L24-28. I don't know if I understand this statement, but SoilFluxPro (Li-COR software) allows to directly upload hundreds of Picarro files simultaneously (up to 2 months). You can choose to open all the files in one single file, and directly merge it with the Li-COR data.

Response: When we first used this import function in SoilFluxPro, we realised that there were problems when importing Picarro files (that are split in one file per hour) by suing the function IMPORT. When a measurement was overlapping two distinct hours (for example a flux estimation from 8:50 am to 9:15 am, the function RECOMPUTE in SoilFluxPro only take into account the first Picarro file. We contacted Licor for this issue and they agree that this was a weakness of the software. They are currently working on it to implement this in a new version of the software. In the meantime, the use of the R function that we developed and that can be found in the supplementary material to merge all Picarro files in one file allows to overcome this issue.

P4 L28-30. One of the best things of using SoilFluxPro is that calculates the fluxes using both linear and exponential fits, which could result in substantial differences in fluxes (see the attached example from my own data). My experience is that exponential equations usually fits better than linear ones (in terms of R2), especially for high flux rates under long chamber closure times.

Response: See responses to previous comments, exponential fits were now used for all flux computations.

P4 L29. Which was the actual length of each measurement without including the dead-band?

Response: See comment from RC2

P5 L5. Why are you not using CO2 measured with Picarro?

Response: Please, see our response to reviewer 1 above (6.).

P5 L7-8. I guess that it has to be the volume of the system (chamber, Li8100, Picarro, multiplexer and tubing). This is really important since the volume of the system is a parameter controlling the minimum detectable flux, so Table 1 might substantially change depending on this "detail".

Response: Yes, it is the volume of the whole system.

P5 L16-17. Again, this can be solved using exponential fits.

Response: See responses below

P6 L7, L13 and L17. I guess these are not the correct figures.

Response: This has been corrected

P6 L14-20. As far as I understood, you kept values higher than MDF (for emissions) and lower than –MDF (for sinks), but what happen with values close in between (-MDF < x < MDF)? What did you do with values close to 0 flux? And what happen if a flux was higher than MDF but had low R2? The same applies for N2O. Which criterion we have to use when measuring gas emissions at low rates? Is it a 0 flux, NA, should we keep the calculated flux regardless of the R2? Choosing one or other criterion might have several implications in order to estimate cumulative or mean fluxes, especially if the data does not have normal distribution and it's not 0 centered. In L11 you describe an R2 criterion for considering stable micrometeorological and chamber conditions based on CO2. Then, why we should apply other criterion for the other gases if the conditions are stable? I understand that for this might not be super relevant for a technical note, but this is a key question if you want to quantify emissions in natural conditions. In my opinion, this is the core of the study, and one of the most challenging issues we need

to address when measuring CH4, N2O and other trace gases. When we have high fluxes, everything is clear, but when we have low fluxes, it turns more complicated. We were discussing this issue in Petrakis et al. 2017, but I still don't have the answers. I guess there is not a silver bullet.

Response: We agree with the reviewer that this is one of the main challenge of CH4 and N2O soil fluxes estimation. For CO2, fluxes from tropical soils are always high so the R2 criterion allows to easily detect measurement issues such as imperfect chamber closure. In this case, it is logical to remove fluxes estimation for the three gases. For CH4 and N2O, we decided (1) to consider fluxes below MDF as null fluxes (i.e. fluxes so small that they are below detection limit) (2) to consider fluxes above MDF but with a low R2 were considered as NA (Not available, fluxes estimation impeded by unknown measurement issues).

Figure 1A. There is something in this panel it's not completely clear. As far as I understand, the air goes from the chamber to the multiplexer, to Li8100, to G2308, to the external pump, to the multiplexer and again into the chamber. However, in the schematic view there is a black circuit (T piece sub-sampling loop) that connects the multiplexer, Li8100, G2308 and the external pump. Since the air composition does not change between these four elements, why the subsampling tub was not inserted in serial at one point of the circuit?

Response: Inserting the subsampling loop in parallel rather in in serial was a proposition from the manufacturer that we followed here.

Table 2. I wonder which closure time did you use in this table (2 or 25min). It would be interesting a comparison between 2 and 25 closure times. I'm not sure you will find differences in the means. This would suggest that short closure times might not affect the annual balance but deviation of the data (as we can see in Ap Figure 2).

Response: In this table, we used all fluxes estimation available after quality checking. Your proposition is interesting but as 2 and 25 minutes estimation were not made on

the same weeks, it is difficult to compare them. We therefore propose to keep this table as it is but we stated in the table caption that this estimation was made using all the data available and we also added the number of data points that were used for this estimation.

Appendix Figure A1. In my opinion, this is one of the most interesting figures in the manuscript and I think it should be place in the main manuscript.

Response: The figure was placed in the main text

Some suggestions: a) Regressions will have better fit if you use exponential equations for estimating the flux. For each flux you can choose linear or exponential depending on the R2.

Response: Exponential fits were now used for all fluxes estimation.

b) Could you display R2 and the coefficients of the regressions between 2 and 25min? Regression B shows a good fit, but it seems that 2 min fluxes tends to overestimate fluxes compared to 25min estimates. Again, this could be an artifact of using linear regressions and not exponential.

Response: This figure has been modified by using exponential fits and the R2 were added

c) It would be interesting plotting the regression for N2O including all values (without removing data using R2 or MDF criterions)? This is related to my comment on Table 2. Apendix Figure A2. Please, edit the figure caption. Petrakis, S., Barba, J., Bond-Lamberty, B. and Vargas, R.: Using greenhouse gas fluxes to define soil functional types, Plant Soil, 1–10, 2017.  

---

## Author Comment (AC5) · 9 Nov 2018

RC3, Referee #3 (Remarks to the Author):

In this manuscript, the authors detail a field-deployed and field-tested system for measuring soil greenhouse gas (GHG) emissions ($CO_2$, $CH_4$ and $N_2O$) from a tropical wet forest; the system leverages a commercially-available automated flux chamber system with a commercially-available CRDS analyzer. More specifically, the authors (a) outline the technical protocol for implementing such a system, (b) report the mean fluxes and variability of $CO_2$, $CH_4$ and $N_2O$ observed over the four-month deployment period and (c) test two chamber closure lengths to determine the most effective experimental design for capturing fluxes above minimum detectable levels. Successfully implementing such a system in the tropics is both difficult and has only been done rarely, so a technical note detailing how to do so is absolutely a contribution to the literature. I have two general/more broad comments about the manuscript, which I detail below, and also include several more specific comments at the end of this review

Response: We thank the reviewer for the positive comments and constructive inputs.

1. Concerns regarding flux calculation protocols In these automated, high-frequency GHG systems, one important set of experimental design and protocol decisions govern how to calculate flux rates and screen for acceptable data points. The authors lay out fairly transparent information about how they calculated their flux rates for each gas, but I wonder if more discussion of the implications of their calculation choices is warranted. I have a few specific questions. Am I correct in understanding that the authors only calculated flux rates for all three gases based on a linear model (Pg 4, Ln 27)? If so, I wonder why they didn't consider also fitting exponential models to the CH4 and N2O fluxes, if not the CO2 fluxes.

Response: We first decided to use linear flux calculations only because we thought that the saturation effects characterised by a plateau after a certain time would be low. However, following your comment and comment from other reviewers, we changed flux calculations in the new version of the manuscript to use exponential estimations.

The authors themselves note on Pg 5, Ln 17 that using certain chamber closure times (which, of course, this paper is very interested in) in combination with a linear flux fit can lead to flux underestimation. Couldn't the "optimal" chamber closure time that the authors attempt to find also include some experimental designs with different closure lengths but non-linear flux fits?

Response: The standard error approach that we used (Nickerson, 2016) is a first order approximation for the MDF from high-frequency measurements and the "true" MDF is a function of the chamber timeseries fit type as well (i.e. Linear, exponential, quadratic).

Nonetheless, while the use of linear regression resulted in systematically smaller fluxes as compared to exponential regression. It is therefore recommended to initially calculate fluxes with linear regression to determine the threshold for "low" fluxes and to recalculate them using exponential regression (Korkiakoski et al., 2017).

Korkiakoski, M., Tuovinen, J.-P., Aurela, M., Koskinen, M., Minkkinen, K., Ojanen, P., Penttilä, T., Rainne, J., Laurila, T. and Lohila, A.: Methane exchange at the peatland forest floor–automatic chamber system exposes the dynamics of small fluxes, 2017.

Nickerson, N.: Evaluating gas emission measurements using Minimum Detectable Flux (MDF), Eosense Inc Dartm. N. S. Can., 2016.

Additionally, the Picarro G2308 records numerous diagnostic variables alongside GHG concentrations, including measures of moisture, temperature, pressure, etc. From methods section 2.6, I am under the impression that fluxes were only struck from the dataset if they were (a) below the MDF, (b) had an R2 for CO2 < 0.9, or (c) only struck N2O data if the SHORT R2 < 0.8. First, the authors might consider including a supplemental figure that justifies their decision not to have data quality rules around CH4, as they do for N2O.

Response: We decided to use the same quality check for CH4 than for N2O and to only struck CH4 data if the SHORT R2 < 0.8. As you can see in the CH4 figure, the main emission or consumption peak during the 2 minutes measurements are still present.

Second, like the other reviewers, I was curious as to how humidity was dealt with, since it appears that the moisture-related diagnostics weren't used to evaluate data quality. Was there a water trap that isn't marked in the instrument set-up diagram?

Response: We regularly checked that no liquid water accumulation occurred in the sampling tubes. Moreover, both analyzers measure also water vapor and its effect on concentration of CO2, CH4 and N2O was accounted for. We also monitored Licor and Picarro inside operating temperature: Picarro temperature remained between 44.99

and 45.01 C and Licor temperature at 51.7 C.

More broadly, as this paper aims to outline best practices for setting up this kind of experiment in the tropics, fleshing out the data management aspect of things would improve the paper, in my opinion.

Response: We think that we already included many aspect of quality check and data management and more information can be found in the new version of the manuscript. All the subsequent figure and analysis were done using R software and we would be happy to share the codes upon request.

2. Technical note vs. data exploration paper The aspects of this paper that serve as a technical note are novel and helpful. That said, the results and discussion section, in which the observed GHG fluxes are analyzed, is perfunctory and is relatively focused on a methodological question: what chamber closure time should be used in this system, and how can others determine what chamber closure time to use in their analogous system? I found myself wishing that there was a more robust analysis of the GHG data itself and the ecological implications of their various findings. See also my comment below about Table 2.

Response: See RC2 (5)

Specific comments: * Pg 7, Ln 30: The authors ultimately recommend a sampling protocol that rotates between short and long closure times. What is preventing them from recommending always doing LONG chamber closures and only using the first two minutes of chamber closure time to calculate the CO2 flux, thus decreasing the amount of human labor needed to swap out the system program once a week?

Response: Setting all chambers as LONG measurements would have led to a maximum of ∼ 3 measurements only per chamber and per day. Mixing LONG and SHORT measurements allows to maximise the number of measurements per chamber and per days while ensuring a reliable estimation of the low N2O fluxes and to capture transient peaks of CH4 and N2O. Moreover, under tropical conditions, a visit once a week is absolutely necessary to ensure a proper maintenance of the setting (removing fallen leaves, branches, etc. . .).

* Pg 7, Ln 23 / Figure A2: This figure is used to justify why LONG chamber readings weren't reliable for N2O flux estimation, but these data don't indicate that the variable fluxes are unreliable, only that they are variable and considerably larger in magnitude than the SHORT N2O flux estimates. Can an additional supplemental figure be added showing the R2 values for the LONG vs. SHORT N2O fluxes? Or some similar figure that shows why the LONG fluxes are considered unacceptable?

Response: If we understood well your question, you are asking why we considered SHORT (not LONG) N2O fluxes as unacceptable. In order to compare the fluxes from SHORT and LONG closure time, we have added a third graph to the Figure 2 (previously supplementary Figure 1). It shows that N2O fluxes estimated using the SHORT closure time are not correlated with fluxes estimated with the LONG closure time (R2 of 0.02) and should not be considered.

* Pg 7, Ln 31: "Our unique system. . ." and Pg 8, Ln 12: "this is the first time that this experimental set up is described and tested under tropical field conditions." I believe an analogous system was described in Puerto Rico (O'Connell et al 2018, Nature Communications, https://doi.org/10.1038/s41467-018-03352-3), though not as a technical or methods paper.

Response: We realised that, although the suggested reference was in the list of references, it was not mentioned in the text and we now cite this reference in the introduction. In this reference, the authors are measuring CO2 and CH4 but not N2O fluxes. Moreover, the description of the experimental design of the automated soil fluxes measurement system in this paper is very short and does not give the specification of the multiplexer not the exact model of the CRDS Picarro analyser that they used. We therefore still believe that our study is the first one describing in detail the simultaneous

measurement of the three GHGs under tropical field conditions.

* Appendix Figure A1: Might be worth including the N2O comparison just as the CO2 comparison is included even though the authors discard the LONG CO2 flux estimates . Response: In the revised manuscript, this figure is now presented in the main text (Figure 2) and also includes the N2O.

* Table 2: A number of the authors' chambers reported mean N2O fluxes below 0. This seems worth mentioning in the results and/or discussion.

Response: We agree with the reviewer and, as suggested, we have added some comments about the respective parts of the N2O and also CH4 emissions and absorptions. We have estimated that 28% of our fluxes indicated a sink for N2O and 72% a source for N2O and that 59% of our fluxes indicated a sink for CH4 and 41% a source for CH4.

---

## Author Comment (AC6) · 9 Nov 2018

SC2, Referee #5 (Remarks to the Author):

This manuscript focused on a very important topic about soil CO2/CH4/N2O fluxes in tropical rainforest. The experiment was well designed. Particularly, this may be the world's first report about in situ and simultaneously measurement of soil CO2/CH4/N2O fluxes at low latitude (between 10ÅęN and 10ÅęS). I would like to give the authors my comments.

Response: Thank you for this positive comment.

1. Important references: To date, through the "Web of Science", I could not find any publication about continuous measurement of soil CO2 efflux (Rs) using the automated chambers in the low latitude tropical forests that between 10_ N and 10_ S. Though two campaign studies in very humid forests (ï'C Âÿs3500 mm of annual precipitation) using automated chambers each in northeastern Australia (17_ S) (Kiese and Butterbach-Bahl, 2002) and northeastern Puerto Rico (18_ N) (Wood et al., 2013) were conducted only less than 6-month period, they observed similar phenomenon with Rs was higher during the dry season but lower during the wet season. Kiese and ButterbachBahl (2002) also measured N2O flux. Conversely, a 4-year continuous measurement of Rs in a seasonal dry (1,250 mm of annual precipitation) tropical forest in western Thailand (14_ N) showed higher Rs in wet season than that of dry season (Hanpattanakit et al., 2015).

Response: In fact, a very recent paper reported continuous monitoring of Rs during three years in the tropical forest of Panama (Rubio and Detto, 2017). Moreover, a previous study conducted at the same site as ours (Paracou site, near the Guyaflux tower) also reported 577 days of Rs measurement (Rowland et al., 2014). Both references highlighted a significant effect of soil moisture on seasonal and diurnal cycles of Rs. Together with the two other references from tropical that you cited, there provide evidences that Rs in tropical forest soils are typically higher in the wet than in the dry season. The other study that you cited (Hanpattanakit et al., 2015) was conducted in a seasonally dry forest which are apparently reacting differently than typical tropical wet forest (precipitations > 2000m/year). Nonetheless, the results that we are presenting in our study were conducted from June to September 2016 which corresponds in our site to the end of the wet season and the onset of the dry season. With these data, we cannot discuss seasonal effects, at least one full year, or more, of measurements would be necessary for this.

Rowland, L., Hill, T. C., Stahl, C., Siebicke, L., Burban, B., Zaragoza‐Castells, J., Ponton, S., Bonal, D., Meir, P. and Williams, M.: Evidence for strong seasonality in the

carbon storage and carbon use efficiency of an Amazonian forest, Glob. Change Biol., 20(3), 979–991, 2014.

Rubio, V. E. and Detto, M.: Spatiotemporal variability of soil respiration in a seasonal tropical forest, Ecol. Evol., 7(17), 7104–7116, 2017.

2. CO2 flux: Empirically, also see the above references, CO2 flux is largely controlled by soil moisture (rain events) at tropical forests. However, based on Fig 3, during 4-month experiment (June-September 2016), most of the chambers did not show temporal variation in CO2 flux. Thus, the authors are suggested to add soil moisture (and temperature) data to Fig 3 and provide some discussion about the (lack of) relationships between Rs and soil moisture and temperature.

Response: As discussed in above, a four months period is limited to go deep into such relationships, especially in tropical forest where temporal and spatial variability of fluxes are high. You can find below a figure that can now be found in the supplementary material of the manuscript displaying the relationship of the three gases with soil moisture. Nonetheless, going deeper in the discussion of the effect of rain event, soil moisture and the relative importance of spatial, seasonal and diurnal variability of fluxes cannot be done with these dataset that was specifically constructed to demonstrate the feasibility of running the system under tropical conditions.

3. CH4 flux: Generally speaking, upland forest soil is a CH4 sink, even lowland tropical forest soil. Compared to Rs, however, CH4 flux is more complex and generally has large spatial variation, because the termite activity can emit CH4 thus offset a partial of the soil CH4 sink. I am confused with Table 2, because ten of the sixteen chambers showed CH4 source. Li-Cor soil chamber (8100-104) can be considered to block most activity of the termite, because the chamber base (collar; 7 cm in height) was inserted _7 cm into the soil and left another 4 cm above the soil; in addition, the chamber has relative additional big metal base surround the collar. On the other hand, inserted chamber base (collar) into the tropical (clay) soil can (sometimes) cause waterlogging

inside the Li-Cor soil chamber (8100-104), which might convert the CH4 sink to CH4 source. Same with CO2 flux, temporal variations in CH4 fluxes also could not be detected in Fig. 4. Also, megascopically, the chambers did not show the common pattern of temporal variation in CH4 fluxes (Fig 4). Sure, this forest has plentiful precipitation (about 3000 mm) and very low elevation, both of these abiotic factors may cause the site as CH4 source. Thus, the authors are suggested to provide some more discussion about (the lack of) spatio-temporal variation in CH4 flux.

Response: Again here, this result can be easily explained by the time frame of the study. Tropical soils are generally considered as sink at a yearly basis but much study show that there are seasonal variation in CH4 fluxes and that tropical soils tend to shift from a sink in the dry season to a sources during the wet season. Here, a four months period is limited to go deep into such relationships, especially in tropical forest where temporal and spatial variability of fluxes are high. You can find below a figure that can now be found in the supplementary material of the manuscript displaying the relationship of the three gases with soil moisture. Nonetheless, going deeper in the discussion of the effect of rain event, soil moisture and the relative importance of spatial, seasonal and diurnal variability of fluxes cannot be done with these dataset that was specifically constructed to demonstrate the feasibility of running the system under tropical conditions.

4. Appendix Figure A1: This figure shows a very general (basic) chamber-problem for measurement of soil GHGs fluxes. Long closure time will cause higher GHGs concentration (if the soil is GHGs source) or lower GHGs concentration (if the soil is GHGs sink) inside the chamber, which will induce underestimation of GHGs flux (saturation effect). Saturation effect is generally positively associated with both flux rate and ratio of the effective chamber volume to the measured soil surface area. Empirically, I believe the 2-mintute closure time is enough for measurement of both CO2 and CH4 flux in tropical forests, even for most temperate and boreal forests. For Li-Cor soil chamber (8100-104), the ratio is (0.0040761/ 0.03178=0.12826 m) = 12.3 cm. However, for

many of the custommade soil chambers, the ratio is generally higher than 12.3 cm, thus this is might be the specific problem (issue) only for Li-Cor soil chamber (8100-104). I suggest the authors feedback this problem to Li-Cor and suggest Li-Cor to draw this problem to their instrument user manual.

Response: Thank you for this feedback. Following comments from the other reviewers, we used exponential fit for estimating all fluxes which improved this saturation issue. Also, as stated in the manuscript, we always used 2 minutes estimation for $CO_2$ fluxes to overcome this issue.

5. Also for Appendix Figure A1: The authors are suggested to re-draw the Appendix Figure A1 indicating different symbols (or color) for each of the four chambers.

Response: Following comments from the other reviewers, this figure has been moved to the main text and now also include $N_2O$. We decided to use different colours (black and grey) for the two distinct weeks that were used for this comparison instead that different colours for the different chambers because it allows a better view of the fact that these two weeks are covering almost the whole range of fluxes that can be encountered in the site.

6. Closure time: When compared Table 1 with Table 2, the closure time of 10 minutes for measurement of N2O flux was enough. Thus, the Table 1 is suggested to be deleted.

Response: We disagree with this comment. A closure time of 10 minutes would have led to a MDF of 0.009 instead of 0.002. In this case, only 82% instead of 96% of the fluxes would have been considered of reliable. We therefore decided to maintain Table 1 in the manuscript as it allows to show that a MDF of 0.002 can only be achieved with a 25 minutes closure time.

7. Additional suggestion 1: To prove the data quality or measurement precision, the authors are suggested to add a plot showing changes in CO2, CH4 and N2O concentrations in the chambers. Following is a sample plot (Sample Fig).

Response: This information has been added.

8. Additional suggestion 2: As I mentioned in the above, this may be the world's first report about in situ and simultaneously measurement of soil $CO_2$/$CH_4$/$N_2O$ fluxes at low latitude (between 10_N and 10_ S). I believe this paper will be a potential high citation rate if the authors can give some more discussion about spatio-temporal variation in $CO_2$/$CH_4$/$N_2O$ fluxes and their control factors. For example, the coefficient of variation (CV) was used to represent the spatial variation. CV of Rs can be calculated by CV = (SD/(mean Rs))_100.

Response: Mean and SD per chambers are available in Table 2 and we added a figure with mean value of each chamber per days for the three gases allowing to visualize the spatio-temporal variability of fluxes.

———————————————————

---

## Author Response (AR2)

RESPONSES TO REVIEWER 2

**Suggestions for revision or reasons for rejection (will be published if the paper is accepted for final publication)**

The authors did a thorough job responding to the reviews and incorporating the comments into the manuscript. I just have two more general comments, and a few minor things to be cleaned up.

*We thank you for agreeing to review the manuscript again and for your positive comments.*

General comments
I still have one question about the flux calculation procedure. You write that all fluxes are calculated using exponential fitting in the Soil Flux pro software. Does that mean you really calculated an exponential flux for each measurement or did you use purely the results from the "Exp Flux" column? The two things are not identical. If the software decides that a linear fit is more suited for a curve, the nonlinear coefficients are set based on the linear fit.

*We recomputed all fluxes using exponential fits and then used the Exp_Flux column. It is clearly stated in Soil Flux pro manual (p. 58) that this value corresponds to exponential fit of the data.*

I know it is not the primary scope of the manuscript, but it would have been nice to see a comment about the effect of the flux calculation on the calculated fluxes. There is a significant difference in the shown flux distributions when you compare Fig. 3 from the revised manuscript with Fig. 2 from the original manuscript. Most notably for N2O. This could eventually impact annual or seasonal balance estimates.

*You can find below a figure presenting the comparison between linear (x-axis) and exponential (y-axis) of the same measurement for all the fluxes. Linear estimation are clearly underestimating fluxes for high fluxes. This figure is now integrated in the supplementary material of the new version of the manuscript.*

[Figure]

Specific comments

Page 1, line 32: Start the sentence with, "After water vapour,…"

*Corrected*

Page 2, line 6: "change" instead of "increase (or decrease)"

*Corrected*

Page 4: Sometimes you write "Li-" instead of "LI-" for the LI-COR instruments". Also "Soil Flux pro" and "Soil flux Pro"

*Corrected throughout the manuscript*

Page 4, line 16: Add "sensors" after "water content"

*Corrected*

Page 5, line 21: At the end of the sentence add "if fluxes are calculated linearly."

*Corrected*

Page 6, line 19: "with previous or following" you mean "adjacent", right?

*You are right. We used adjacent in the new version of the manuscript.*

Page 6, line 34: "varied" instead of "varies"

*Corrected*

Page 7, line 11: "could" instead of "can" ; the average flux value and the uncertainty value should have the same decimal

*Corrected*

Page 7, line 15: one redundant full stop

*Corrected*

Page 8, line 1: "that" instead of "than"

*Corrected*

Page 8, line 5: a comma or "and" missing before "(4)"; "could" instead of "can" (same in the N2O section);

*Corrected*

Page 8, line 20: "displayed" instead of "display"

*Corrected*

The conclusions can be shortened. There are some repetitions in it.

*The conclusion has been shortened.*

Figure 6: In contrast to what the caption says, y-axis don't have the same limits.

*Corrected*